# SCALING THE PRIOR: SIZE-CONSISTENT GEOMETRIC DIFFUSION FOR 3D MOLECULAR GENERATION

## ABSTRACT

Diffusion models usually operate in fixed-dimensional metric spaces. In contrast, geometric molecular data naturally vary in dimensionality as molecules have different sizes (numbers of atoms). As a simple adaptation, existing diffusion models for geometric molecular generation employ network architectures that can handle variable-sized inputs, such as graph neural networks and transformers. **However, these approaches overlook the fact that the molecular size also determines the spatial scale of the atomic coordinates, which in turn induces inconsistent behaviors in the generative trajectories across different molecular sizes.** The generative process of geometric diffusion for 3D molecular generation can be viewed as first establishing a coarse structural target, followed by progressively refining the precise atomic positions. In particular, larger molecules tend to establish coarse structures earlier than smaller molecules due to their larger spatial scales relative to that of the noise. As a result, the reverse process becomes inconsistent across molecular sizes, with the denoising trajectories relying heavily on molecular sizes rather than on a unified generative pattern. In this work, we are the first to identify and analyze this size-induced inconsistency through a decomposition of the denoising dynamics, which reveals how spatial scale affects the progression of molecular formation, in both 3D structures and atom types. Building on this insight, we propose Scaling the Prior (StP), a simple yet effective approach that normalizes the learning and generative process across molecular sizes by rescaling the prior distribution based on molecular sizes. This adjustment harmonizes the denoising trajectories, enabling the model to learn a unified generative pattern and produce consistently high-quality molecules.

## 1 INTRODUCTION

In recent years, diffusion models (Ho et al., 2020; Song et al., 2020; 2021) have achieved great advances in both theory and practical applications (Nichol & Dhariwal, 2021; Rombach et al., 2022; Nie et al., 2025; Zhang et al., 2023). These models consist of a forward process and a reverse process. The forward process is a fixed Markov chain that incrementally corrupts data samples by adding noise until the original signal is lost. The reverse process is modeled by a denoising neural network, which is trained to iteratively undo the corruption and reconstruct the original data; once trained, new samples are generated by running the reverse process, beginning from pure Gaussian noise. Motivated by their success in vision tasks such as image generation, researchers have extended diffusion models to various domains, including 3D molecular generation (Hoogeboom et al., 2022; Xu et al., 2023). In particular, 3D molecular generation includes generating both the continuous 3D atomic configurations as well as discrete categorical features such as atom types. Diffusion models usually operate in fixed-dimensional metric spaces, such as $\mathbb{R}^{H \times W \times C}$ for images, where $H$, $W$, and $C$ are the height, width, and number of channels, respectively. However, 3D molecular data varies in dimensionality because molecules have different numbers of atoms. A straightforward strategy taken by existing work is to adopt network architectures that are inherently suited to variable-sized[1] inputs, such as graph neural networks (GNNs) and transformers (Hoogeboom et al., 2022; Ding & Hofmann, 2025). However, 3D molecular data presents a unique challenge that has yet to be fully identified and addressed.

---

[1]In this work, size always denotes the number of atoms; the spatial extent of a molecule is called *scale*.

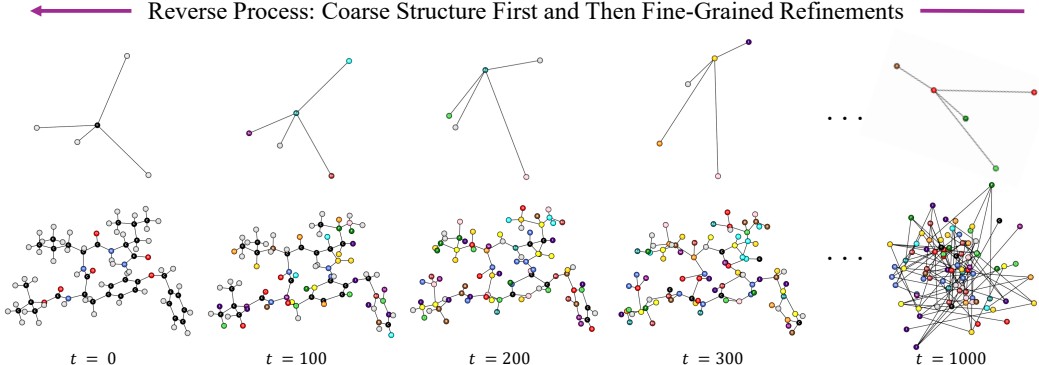

Figure 1: The diffusion/forward process of two 3D molecules of varying sizes from the GEOM-Drugs dataset (Axelrod & Gomez-Bombarelli, 2022), using the common noise schedule from prior work (Hoogeboom et al., 2022). Colors represent different atom types. **Existing works apply the same Gaussian prior across molecules of all sizes. However, it leads to size-induced inconsistencies.** The geometry of the 5-atom molecule on the top is already unrecognizable after 200 steps, while the 91-atom molecule on the bottom still preserves its overall geometry. **This size-induced inconsistency makes larger molecules stabilize earlier in the reverse process and show better performance**, as shown in Sec. 3.1 and Sec. 3.2. Atom types are represented as one-hot vectors with the same scale across molecular sizes. However, inconsistencies in spatial scales also propagate to atom types as shown in Sec. 3.2. These two molecules differ greatly in their spatial scales; they are drawn to the same scale for visualization purposes.

**The generative process in 3D molecular diffusion can be viewed as first establishing a coarse structural target, followed by progressively refining atomic positions.** This challenge arises from a distinctive property of molecular data; that is, molecules vary in size (number of atoms), which naturally induces different spatial scales. In contrast, pixel values in image data are always 0 to 255, independent of the image dimension. For smaller molecules, the spatial extent is small, so the same level of noise corresponds to larger perturbations on a relative scale compared to larger molecules. This discrepancy leads to fundamentally different denoising behaviors across molecular sizes. For larger molecules, the generative process establishes a coarse structural target and begins fine-grained positional adjustments early, while for small molecules, at the same reverse time step, it remains focused on forming the coarse structures. An illustration is shown in Fig. 1. **Therefore, the generative process behaves inconsistently across molecular sizes**, leading to suboptimal performance.

In this work, we analyze denoising dynamics through a decomposition of shape and atom types, which separates the generative process into a shape-preserving radial part of the 3D configuration and a categorical component governing atom types. This decomposition provides a quantitative lens to reveal the scale-dependent inconsistency, showing how **molecules of different sizes establish their coarse structural targets at different rates during the reverse process**. As we aim to mitigate this issue, a natural question arises: *Can we simply normalize the data so that the 3D atomic configurations of molecules of different sizes have the same scale?* Unfortunately, directly normalizing the molecular data would distort important chemical information such as bond lengths, which encode the type and strength of chemical interactions (Kindermans & Müller, 2018; Qu et al., 2025). We provide more discussion in Sec. 3.3 and ablation studies in Sec. 4.4. Based on these insights, we propose a simple yet effective solution: Scaling the Prior (StP). Instead of rescaling the molecular data itself, which would irreparably distort chemically meaningful structural information, StP rescales the prior distribution based on molecular sizes. **By aligning the noise scale with the spatial extent of a molecule, this adjustment harmonizes the denoising trajectories across different molecular sizes.** As a result, the model learns a unified generative pattern, establishing structural target and refining local details in a consistent manner regardless of size.

**In summary, our contribution can be summarized as follows:** ① We are the first to identify and introduce an important, yet often overlooked, size-dependent inconsistency in diffusion-based 3D molecular generation. ② We propose principled metrics to quantify the structural and atom type changes in the diffusion process, which can provide future insights. ③ We provide a simple yet effective solution, StP, which rescales the noise based on the molecular size, harmonizing denoising

trajectories across molecule sizes while preserving chemically meaningful structural information. ④ Experimental results demonstrate that StP significantly improves validity, stability, and overall quality of generated molecules, without introducing additional architectural complexity.

## 2 PRELIMINARIES AND RELATED WORK

### 2.1 DIFFUSION MODELS

Diffusion models learn a data distribution by inverting a forward diffusion process. Given a data point $\boldsymbol{x}$, the forward diffusion process adds increasing levels of Gaussian noise to it:

$$q\left(\boldsymbol{z}_t \mid \boldsymbol{x}\right) = \mathcal{N}\left(\boldsymbol{z}_t \mid \alpha_t \boldsymbol{x}, \sigma_t^2 \boldsymbol{I}\right), \ t = 0, 1, 2, \ldots, T, \tag{1}$$

where $\boldsymbol{z}_t$ denotes a noisy version of $\boldsymbol{x}$ at time $t$, $\alpha_t > 0$ controls how much of the original signal is retained, and $\sigma_t > 0$ controls how much noise is injected. A special case is the variance preserving process in which $\alpha_t^2 + \sigma_t^2 = 1$ (Ho et al., 2020; Song et al., 2021). In general, $\alpha_t \approx 1$ at $t = 0$ and then monotonically decreases to 0 at $t = T$, corresponding to the progressive corruption of the data into pure noise. The diffusion process is Markovian and can equivalently be expressed by its transition distributions:

$$q\left(\boldsymbol{z}_t \mid \boldsymbol{z}_s\right) = \mathcal{N}\left(\boldsymbol{z}_t \mid \alpha_{t|s} \boldsymbol{z}_s, \sigma_{t|s}^2 \boldsymbol{I}\right), \tag{2}$$

where $T \geq t > s \geq 0$, $\alpha_{t|s} = \alpha_t / \alpha_s$, and $\sigma_{t|s}^2 = \sigma_t^2 - \alpha_{t|s}^2 \sigma_s^2$. Given equation 1 and equation 2, we can derive the posterior of the transitions conditioned on $\boldsymbol{x}$:

$$q\left(\boldsymbol{z}_s \mid \boldsymbol{z}_t, \boldsymbol{x}\right) = \mathcal{N}\left(\boldsymbol{z}_s \mid \boldsymbol{\mu}_{s|t}\left(\boldsymbol{z}_t, \boldsymbol{x}\right), \sigma_{s|t}^2 \boldsymbol{I}\right), \tag{3}$$

where $\boldsymbol{\mu}_{s|t}\left(\boldsymbol{z}_t, \boldsymbol{x}\right)$ and $\sigma_{s|t}^2$ can be derived analytically:

$$\boldsymbol{\mu}_{s|t}\left(\boldsymbol{z}_t, \boldsymbol{x}\right) = \frac{\alpha_{t|s} \sigma_s^2}{\sigma_t^2} \boldsymbol{z}_t + \frac{\alpha_s \sigma_{t|s}^2}{\sigma_t^2} \boldsymbol{x}, \quad \sigma_{s|t}^2 = \frac{\sigma_{t|s}^2 \sigma_s^2}{\sigma_t^2}. \tag{4}$$

The posterior distribution defines the reverse process (the generative process). The posterior distribution is generally unknown in the generative process as it is conditioned on unknown $\boldsymbol{x}$. In practice, a neural network parameterization $\phi$ is trained to approximate the true posterior:

$$p_\phi\left(\boldsymbol{z}_s \mid \boldsymbol{z}_t\right) = q\left(\boldsymbol{z}_s \mid \boldsymbol{z}_t, \boldsymbol{x}_\phi\left(\boldsymbol{z}_t, t\right)\right). \tag{5}$$

A variational lower bound on the log-likelihood of $\boldsymbol{x}$ can be derived for the diffusion model as:

$$\log p(\boldsymbol{x}) \geq \mathcal{L}_0 + \mathcal{L}_{\text{prior}} + \sum_{t=1}^{T} \mathcal{L}_t, \tag{6}$$

where $\mathcal{L}_0 = \log p\left(\boldsymbol{x} \mid \boldsymbol{z}_0\right)$ models the reconstruction error, $\mathcal{L}_{\text{prior}} = -\text{KL}\left(q\left(\boldsymbol{z}_T \mid \boldsymbol{x}\right) \mid p\left(\boldsymbol{z}_T\right)\right)$ models the divergence between the prior standard normal distribution and the final latent distribution $q\left(\boldsymbol{z}_T \mid \boldsymbol{x}\right)$, and $\mathcal{L}_t = -\text{KL}\left(q\left(\boldsymbol{z}_s \mid \boldsymbol{x}, \boldsymbol{z}_t\right) \mid p\left(\boldsymbol{z}_s \mid \boldsymbol{z}_t\right)\right)$ quantifies the divergence between the learned posterior distribution and the true posterior distribution. Following prior work (Ho et al., 2020), instead of directly predicting $\boldsymbol{x}$, optimization is easier when predicting the noise instead. In particular, $\boldsymbol{z}_t = \alpha_t \boldsymbol{x} + \sigma_t \boldsymbol{\epsilon}$, $\boldsymbol{\epsilon} \sim \mathcal{N}\left(0, \sigma_t^2 \boldsymbol{I}\right)$, then the neural network $\phi$ outputs $\hat{\boldsymbol{\epsilon}} = \phi\left(\boldsymbol{z}_t, t\right)$, so that $\hat{\boldsymbol{x}} = \left(1/\alpha_t\right) \boldsymbol{z}_t - \left(\sigma_t/\alpha_t\right) \hat{\boldsymbol{\epsilon}}$. Let $\text{SNR}(t)$ be the *signal-to-noise ratio* (Kingma et al., 2023), defined as $\text{SNR}(t) = \alpha_t^2 / \sigma_t^2$, the term $\mathcal{L}_t$ can be further expanded as:

$$\mathcal{L}_t = \mathbb{E}_{\boldsymbol{\epsilon} \sim \mathcal{N}(\boldsymbol{0}, \boldsymbol{I})} \left[ \frac{1}{2} \left( \frac{\text{SNR}(t-1)}{\text{SNR}(t)} - 1 \right) \|\boldsymbol{\epsilon} - \hat{\boldsymbol{\epsilon}}\|^2 \right]. \tag{7}$$

### 2.2 3D MOLECULAR DIFFUSION

We are interested in generating 3D molecules from scratch. Specifically, a molecule with $N$ atoms can be represented as $(\boldsymbol{x}, \boldsymbol{h})$, where $\boldsymbol{x} = (\boldsymbol{x}_1, \boldsymbol{x}_2, \ldots, \boldsymbol{x}_N) \in \mathbb{R}^{N \times 3}$ corresponds to the atomic coordinates in the 3D space and $\boldsymbol{h} = (\boldsymbol{h}_1, \boldsymbol{h}_2, \ldots, \boldsymbol{h}_N) \in \mathbb{R}^{N \times d}$ corresponds to $d$-dimensional features of the atoms, e.g., one-hot encoding of atom types. Pioneering works in 3D molecular generation with diffusion models, such as EDM (Hoogeboom et al., 2022) and GeoLDM (Xu et al., 2023),

emphasize the importance of $E(3)$-invariance of the molecular distribution: Euclidean transformations (translations and rotations) do not change the underlying molecule, and thus the probability density should remain unchanged. Translation invariance is enforced by operating in a zero-mean linear subspace, in which the coordinates of both molecules and noise are centered at the origin: $\boldsymbol{x} := (\boldsymbol{x}_1 - \bar{\boldsymbol{x}}, \boldsymbol{x}_2 - \bar{\boldsymbol{x}}, \ldots, \boldsymbol{x}_N - \bar{\boldsymbol{x}}) \in \mathbb{R}^{(N-1) \times 3}$, where $\bar{\boldsymbol{x}} = \frac{1}{N} \sum_{i=1}^{N} \boldsymbol{x}_i = \boldsymbol{0}$. Rotation invariance is further ensured by employing a denoising network that is equivariant to rotations (Hoogeboom et al., 2022; Xu et al., 2022). Nevertheless, recent studies suggest that rotation equivariance is not always necessary (Ding & Hofmann, 2025; Joshi et al., 2025), and that employing equivariant denoising networks may introduce substantial computational overhead. Besides diffusion models, flow-based approaches have also attracted significant attention (Dunn & Koes, 2024; Irwin et al., 2025; Song et al., 2023; 2024; Hong et al., 2025). These methods typically employ a Gaussian prior over atomic positions, and it has been shown that, under a Gaussian prior, they are mathematically equivalent to diffusion models (Albergo et al., 2023; Ma et al., 2024; Gao et al., 2025).

The theory of diffusion models and their stochastic differential equation (SDE) formulation are defined over fixed-dimensional metric spaces. This creates a fundamental mismatch for molecules, whose sizes vary and thus cannot be naturally embedded into a single fixed ambient space. In practice, a workaround is to employ denoising networks that can work with variable-sized inputs, such as GNNs and transformers (Hoogeboom et al., 2022; Xu et al., 2023; Joshi et al., 2025; Ding & Hofmann, 2025). One way to view this workaround is through the lens of amortization: there is a diffusion process for molecules of each size; however, a unified denoising network is trained to amortize across these processes, learning to handle variable molecular sizes within a single model. In this view, the SDE framework remains conceptually defined for a fixed-dimensional ambient space, but the denoising network provides a shared parametrization that generalizes across different molecule sizes. While this amortization enables the treatment of variable-size data at the architectural level, it overlooks another critical factor: Molecular size also affects the spatial scale of the coordinates, which in turn leads to inconsistencies in the diffusion process.

## 3 SCALING THE PRIOR: SIZE-CONSISTENT GEOMETRIC DIFFUSION

### 3.1 OBSERVING SIZE-INDUCED INCONSISTENCIES

To begin, we present an intriguing and counterintuitive phenomenon in 3D molecular generation with diffusion models. **As shown in Fig. 2, there is a clear trend that as molecular size increases, sampling quality increases, despite the fact that large molecules have higher structural complexity and lower data availability.** Similar inconsistencies can be observed in various diffusion-based 3D molecular generation methods and datasets; we present the experimental details and additional results in Appendix A.

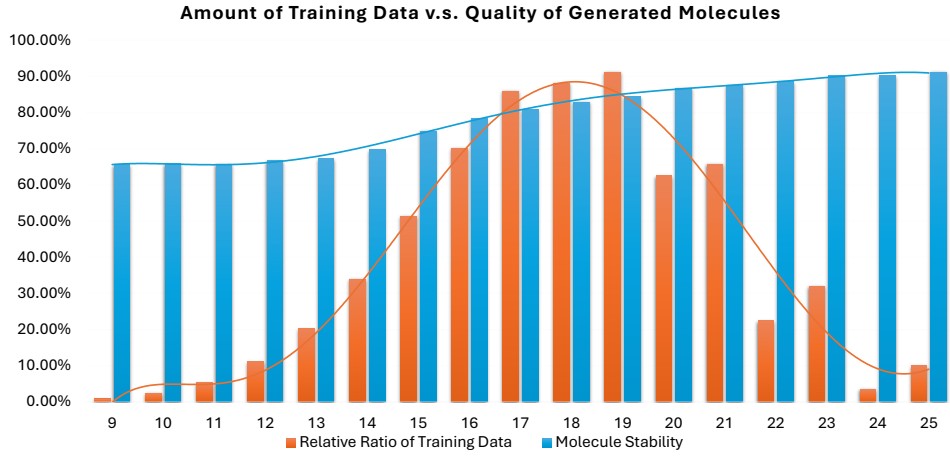

Figure 2: Sampling quality ($y$-axis) versus molecular size ($x$-axis) with EDM (Hoogeboom et al., 2022) on the QM9 dataset (Ramakrishnan et al., 2014). Clearly, larger molecules achieve higher sampling quality despite the scarcity of corresponding training samples. For visualization, the relative ratio of training data for the molecular size with the most training samples is normalized to the highest stability value, and all other sizes are scaled relative to this.

This phenomenon calls for careful analysis to uncover its underlying causes and implications in diffusion-based 3D molecular generation. Therefore, we delve deeper to uncover its underlying cause. In Sec. 3.2, we identify the root cause from the unique characteristic of 3D molecular data that different sizes of molecules have different scales, which leads to inconsistencies in the generative process and generated sample qualities. Building on this analysis, in Sec. 3.3, we propose a solution tailored to mitigate the observed inconsistencies and improve generative performance across molecular sizes.

**Discussion on Latent Diffusion Models.** Latent diffusion models encode the coordinates and atom features $(\boldsymbol{x}, \boldsymbol{h})$ into a lower-dimensional latent space. Existing latent diffusion models, e.g., GeoLDM and RADM, still separate coordinates and atom features, keeping coordinates in the 3D Euclidean space in the latent representation as $\boldsymbol{x'} \in \mathbb{R}^{(N-1)\times 3}$, $\boldsymbol{h'} \in \mathbb{R}^{N \times d'}$, where typically $d' < d$. Most importantly, coordinates rarely change in the latent space (up to $SO(3)$ transformations for RADM), and size-induced inconsistencies remain in these latent diffusion models. The analysis and techniques presented in this work are still applicable. A more detailed discussion and concrete examples are provided in Appendix B.

## 3.2 Unveiling Size-Induced Inconsistencies

Diffusion models can be viewed as an approximate autoregression in the frequency domain (Rissanen et al., 2023). In the forward process, noise gradually destroys high-frequency details while low-frequency overall structures persist longer. In generation, the model first constructs the coarse structure and then progressively adds higher-frequency details towards the generation target.

*Remark* 3.1. The generation target refers to all possible clean data samples that can be inferred from noisy samples. For example, given a mildly noisy image, we may infer its cleaned form.

Importantly, the generation target is not always a single clean data point. For example, at $t = T$, pure noise provides no information about any specific clean sample; the generation target is ambiguous and can contain any data sample in the distribution. In contrast, at values of $t$ closer to 0, the slightly noised input corresponds only to a limited subset of similar data samples (e.g., molecules with similar 3D configurations). **As $t$ goes to $0$, the noise level decreases, and the generation target becomes more precise and eventually converges to a concrete clean data point (e.g., a specific molecule).**

To this end, the generation target of a noisy sample $(\boldsymbol{z}_t^x, \boldsymbol{z}_t^h)$, which contains both noisy coordinates and atom types, can be defined as

$$\text{Target}\left(\boldsymbol{z}_t^x, \boldsymbol{z}_t^h\right) = \left\{ (\boldsymbol{x}, \boldsymbol{h}) \;\middle|\; p_\phi(\boldsymbol{x}, \boldsymbol{h} \mid \boldsymbol{z}_t^x, \boldsymbol{z}_t^h) \geq \tau \cdot \max_{\boldsymbol{x'}, \boldsymbol{h'}} p_\phi(\boldsymbol{x'}, \boldsymbol{h'}, \mid \boldsymbol{z}_t^x, \boldsymbol{z}_t^h) \right\}, \quad (8)$$

where $0 < \tau \leq 1$ is a threshold parameter. Intuitively, in this definition, the generation target corresponds to the clean data samples at $t = 0$ whose likelihood remains within a fraction $\tau$ of the most probable sample under the posterior (the noisy sample).

For molecular diffusion, the generative process first finds a coarse atomic structure and then refines the precise atomic positions toward a concrete molecule as the generation target becomes more precise. However, we notice that there is an inconsistency across molecular sizes: For larger molecules, the generation target becomes precise faster, and the generative process initiates fine-grained positional adjustments toward concrete molecules earlier. In contrast, for smaller molecules at the same reverse time step, it still continues to focus on forming the coarse structures. Although directly comparing the generation target in equation 8 for different time $t$ is intractable; we decompose it into two generation target alignment components, the *shape alignment coefficient* $\gamma_t$ and the *atom type alignment ratio* $\beta_t$, to quantify this phenomenon. Concretely, given a noisy sample $\boldsymbol{z}_t = (\boldsymbol{z}_t^x, \boldsymbol{z}_t^h)$, we estimate a representative of $\text{Target}(\boldsymbol{z}_t^x, \boldsymbol{z}_t^h)$ as $(\hat{\boldsymbol{x}}_t, \hat{\boldsymbol{h}}_t) = (1/\alpha_t)\,\boldsymbol{z}_t - (\sigma_t/\alpha_t)\,\hat{\boldsymbol{\epsilon}}^2$, which is the predicted clean data at time $t$. For a generative trajectory $\{\boldsymbol{z}_t\}_{t=0}^T$, we take $(\hat{\boldsymbol{x}}_0, \hat{\boldsymbol{h}}_0)$ as its final generation target. By comparing the predicted clean data at time $t$: $(\hat{\boldsymbol{x}}_t, \hat{\boldsymbol{h}}_t)$ with the predicted clean data at time 0: $(\hat{\boldsymbol{x}}_0, \hat{\boldsymbol{h}}_0)$, we can assess the rate at which the generation target converges to a concrete

---

[2]Following prior work, the 3D coordinates and atom features are concatenated, which are then processed by a single denoising network to produce the combined noise prediction.

molecule with respect to time $t$. The convergence in generation target is quantified by:

$$\gamma_t = \frac{\langle \hat{\boldsymbol{x}}_t, \hat{\boldsymbol{x}}_0 \rangle}{\|\hat{\boldsymbol{x}}_0\|^2}, \quad \beta_t = \frac{1}{N} \sum_{n=1}^{N} \mathbf{1} \left[ \hat{\boldsymbol{h}}_{t,n} = \hat{\boldsymbol{h}}_{0,n} \right], \tag{9}$$

where $\mathbf{1}[\cdot]$ is the indicator function checking if the $n$-th atom types are the same, $\gamma_t$ is the radial coefficient indicating the alignment of the shape between $\hat{x}_t$ and $\hat{x}_0$ (how much of $\hat{x}_t$ lies in the direction of $\hat{x}_0$), and $\beta_t$ is the ratio of atoms whose types remain unchanged. We adopt the radial projection to quantify the convergence of generation target in atomic coordinates because the Euclidean norm difference $|\hat{\boldsymbol{x}}_t - \hat{\boldsymbol{x}}_0|$ is itself sensitive to the data scale, while the radial projection emphasizes directional alignment and relative magnitude consistency. **Overall, $\gamma_t$ and $\beta_t$ reflect how quickly the denoising trajectory "lines up" with the final 3D geometric structure and atom types, respectively.**

Figure 3: Inconsistent convergence rates across molecule sizes on QM9 with EDM. Values closer to 1 indicate better alignment with the final generation target. Larger molecules tend to stabilize more quickly than smaller ones, resulting in discrepancies in generation behavior. It is important to note that atom types are represented as one-hot vectors, which remain on the same scale across different molecule sizes; however, they are still affected by the stabilization of the 3D geometry.

By construction, $\gamma_t$ and $\beta_t$ provide complementary views of the convergence of the generation target. As the reverse diffusion proceeds toward $t \to 0$, both $\gamma_t$ and $\beta_t$ should approach 1. We now track these two quantities, $\gamma_t$ and $\beta_t$, across molecules of different sizes to study how generation target emerges during generation. As shown in Fig. 3, there is a clear difference in the rate of convergence between different molecular sizes; **as the molecular size grows, both $\gamma_t$ and $\beta_t$ saturate earlier in the reverse trajectory.** Details and additional results on GEOM-Drugs and with other backbone diffusion models are shown in Appendix A.2. This indicates that larger molecules quickly stabilize to a coarse generation target, while smaller molecules continue to undergo structural adjustments for longer. Such inconsistencies have significant implications for molecular generation quality.

### 3.3 SCALING THE PRIOR: HARMONIZING GENERATIVE DYNAMICS

To mitigate the size inconsistency in the generative process, a straightforward approach is to ensure that molecules of different sizes are on the same scale. Atom types are categorical features and, in principle, share the same scale across molecular sizes. However, we still observe systematic differences because atom-type predictions are *not independent* of coordinates. Both modalities are processed simultaneously by the same denoising network. As a result, the varying scales of coordinates across molecular sizes are the main issue. To this end, we propose to Scale the Prior. Instead of using a unified standard Gaussian as the prior for coordinates, we introduce a size-dependent variance for the prior. Specifically, the marginal distribution in the forward process becomes:

$$q_\gamma \left( \boldsymbol{z}_t^x \mid \boldsymbol{x} \right) = \mathcal{N} \left( \boldsymbol{z}_t^x \mid \alpha_t \boldsymbol{x}, \gamma_N^2 \sigma_t^2 \boldsymbol{I} \right),$$
$$q \left( \boldsymbol{z}_t^h \mid \boldsymbol{h} \right) = \mathcal{N} \left( \boldsymbol{z}_t^h \mid \alpha_t \boldsymbol{h}, \sigma_t^2 \boldsymbol{I} \right), \tag{10}$$

where $q_\gamma$ denotes the marginal distribution from StP to be distinguished from that of the standard diffusion in equation 1, $\gamma_N$ is a size-varying scaling factor, and $\sigma_t$ is from the original noise schedule as in equation 1. For clarity, we write the forward process separately for 3D positions and atom features; however, since the Gaussian is isotropic, this is equivalent to writing them jointly.

**Proposition 3.2.** *The marginal distributions at any time $t$ of Scaling the Prior are equivalent to those of normalizing the data $\boldsymbol{x}$ by a factor of $\frac{1}{\gamma_N}$ before the forward process, and subsequently unnormalizing after sampling.*

Proposition 3.2 is straightforward to prove; nevertheless, we include the proof in Appendix C. This result establishes a direct connection between StP and normalization. In point cloud diffusion in computer vision, the data is often already approximately on the same scale (Luo & Hu, 2021), or per-sample normalization is applied (Tyszkiewicz et al., 2023). Here, per-sample normalization refers to schemes that are not consistent across the dataset; for example, normalizing molecules differently depending on their sizes. **A key distinction between StP and standard normalization is that StP preserves structural information, such as bond lengths, consistently across the dataset, whereas per-sample normalization does not. Structural information, such as bond lengths, is fundamental to molecular modeling (Kindermans & Müller, 2018; Qu et al., 2025).** Naive per-sample normalization can alter chemically important distances, leading to generative models that struggle to capture consistent and chemically meaningful geometry. Concretely, in StP, the forward process in equation 10 scales the entire coordinate space by $\alpha_t$ and injects isotropic Gaussian noise. As a result, the expected interatomic distances, including bond lengths, are $\mathbb{E}\left[|z_{t,i}^x - z_{t,j}^x|\right] = \alpha_t |x_i - x_j|$ ($i, j$ denote the $i$-th and $j$-th atom, respectively), which shows that the expectation of bond lengths is purely determined by the global scaling factor $\alpha_t$ and remains consistent across time steps. On contrast, in per-sample normalization, the expected interatomic distances $\mathbb{E}\left[|z_{t,i}^x - z_{t,j}^x|\right] = \frac{\alpha_t}{\gamma_N}|x_i - x_j|$, which introduces an arbitrary size-dependent factor $\gamma_N$, altering the expected bond geometry. In addition, we provide a learning perspective of StP in Appendix A.3, which offers a high-level illustration of why StP is beneficial for learning the underlying molecular distribution.

**Normalizing the Scales.** Since it is now clear that StP is equivalent to normalization but preserves chemically meaningful structural information, we can define $\gamma_N$ so that it effectively normalizes molecules of different sizes to the same scale. For a 3D configuration $\boldsymbol{x} = (\boldsymbol{x}_1, \boldsymbol{x}_2, \ldots, \boldsymbol{x}_N) \in \mathbb{R}^{N \times 3}$, assuming zero-mean as described in Sec. 2.2, we define the *scale* of $\boldsymbol{x}$ as:

$$s(\boldsymbol{x}) = \frac{1}{|\operatorname{Hull}(\boldsymbol{x})|} \sum_{v \in \operatorname{Hull}(\boldsymbol{x})} \|v\|_2, \tag{11}$$

where $\operatorname{Hull}(\boldsymbol{x})$ denotes the set of vertices (atoms) of the convex hull of $\boldsymbol{x}$ and $|\operatorname{Hull}(\boldsymbol{x})|$ is the number of convex hull vertices. For reference, we provide the definition of convex hull in Appendix A.4. To obtain a normalization factor that is consistent across molecules of different sizes, for molecules of size $N$, we define the *size-specific scale* as the average scale over all the molecules of size $N$. We now can define $\gamma_N$ as the in proportion to the size-specific scale. Mathematically:

$$\gamma_N = \frac{\mathbb{E}_{\boldsymbol{x} \sim \mathcal{D}_N}[s(\boldsymbol{x})]}{Z}, \tag{12}$$

where $\mathcal{D}_N$ is the subset of molecules containing exactly $N$ atoms in the training set, and $Z$ is a normalization factor that is the same in all molecular sizes and will not affect the relative scale across different sizes. An empirical choice can be $Z = \mathbb{E}_{\boldsymbol{x} \sim \mathcal{D}}[s(\boldsymbol{x})]$ with $\mathcal{D}$ being the entire dataset.

## 4 EXPERIMENTS

### 4.1 EXPERIMENTAL SETUP

**Datasets.** We evaluate our approach on the QM9 (Ramakrishnan et al., 2014) and GEOM-Drugs (Axelrod & Gomez-Bombarelli, 2022) benchmarks, which are standard and widely used benchmarks in 3D molecular generation. The QM9 dataset contains 130K molecules with up to 29 atoms (including hydrogens). The GEOM-Drugs dataset comprises 430K molecules with up to 181 atoms and an average size of $44.4$ atoms. For both datasets, we follow the exact data splits and setup used in Hoogeboom et al. (2022).

**Baselines.** We use EDM, RADM (DiT-B), and GeoLDM, which are state-of-the-art 3D molecular diffusion models, as backbone models for our proposed StP and compare with the original without StP. To ensure a fair comparison, we strictly follow the implementation details provided in the original works, except that we might use a different batch size due to computational constraints. We also provide results for several **non-diffusion-based** 3D molecular generation models for reference, including G-SchNet (Gebauer et al., 2019), ENF (Satorras et al., 2021), EDM-bridge (Wu et al., 2022), EquiFM (Song et al., 2023), and GeoBFN (Song et al., 2024). All diffusion and flow baselines are reported with $1,000$ steps. GeoBFN also has a $2,000$-step variant; we report the $1,000$-step version for consistency. The baseline results are directly obtained from their original works.

**Evaluation Metrics.** Following prior work, the bond types are based on the pairwise atomic distance and the atom types. We report the following evaluation metrics: (1) *Atom Stability*: the percentage of atoms in all generated molecules with the correct valence; (2) *Molecule Stability*: the percentage of generated molecules in which all atoms are stable; and (3) *Validity×Uniqueness* of the generated molecules as measured by RDKit. For the GEOM-Drugs dataset, following prior work, we only report atom stability and validity, since molecule stability is close to 0 and uniqueness is close to 1 for all methods. Following pioneering works in 3D molecular generation (Hoogeboom et al., 2022; Xu et al., 2023), we do not use explicit bond information, and we do not apply any post-hoc refinement using computational chemistry software (e.g., Open Babel) to improve the generated molecules. There are works that explicitly use bond information or apply post-hoc refinements (Irwin et al., 2025; Dunn & Koes, 2025); these approaches naturally achieve better reported performance. Therefore, our work is not directly comparable to these works. Following prior work, for all experiments in this section, $10,000$ molecules are sampled, **the experiments are repeated** $3$ **times with the mean values reported**, and the standard division is not reported if it is negligible.

## 4.2 StP: Improved Generation Qualities

Table 1: Results of different 3D molecular generation pipelines. Three SOTA 3D molecular diffusion models, EDM, GeoLDM, and RADM, are adapted into their StP (our method) variants. The standard deviations are reported for QM9 after $\pm$; they are negligible after rounding for GEOM-Drugs. Whenever an StP variant surpasses its original model, the result is highlighted in **orange**. The overall best performing model is highlighted in **red**.

| | QM9 | | | | GEOM-Drugs | |
|---|---|---|---|---|---|---|
| | Atom Stab (%) | Molecule Stab (%) | Valid (%) | Valid× Unique (%) | Atom Stab (%) | Valid (%) |
| Dataset | 99.00 | 95.20 | 97.70 | 97.70 | 86.50 | 99.90 |
| G-SchNet | 95.70 | 68.10 | 85.50 | 80.30 | - | - |
| ENF | 85.00 | 84.90 | 40.20 | 39.40 | - | - |
| EDM-bridge | 98.80 | 84.60 | 92.00 | 90.70 | 82.40 | 92.80 |
| EquiFM | 98.90 | 88.30 | 94.70 | 93.50 | 84.10 | 98.90 |
| GeoBFN | 99.08 | 90.87 | 95.31 | 92.96 | 85.60 | 92.08 |
| EDM | 98.70 | 82.00 | 91.90 | 90.70 | 81.30 | 92.60 |
| EDM-StP | **98.83±0.03** | **88.07±0.22** | **94.41±0.08** | **92.63±0.14** | **84.11** | **95.59** |
| RADM | 98.50 | 87.30 | 94.10 | 91.70 | 85.00 | 99.30 |
| RADM-StP | **98.59±0.01** | **87.62±0.10** | **94.19±0.17** | 91.51±0.15 | **85.27** | **99.49** |
| GeoLDM | 98.90 | 89.40 | 93.80 | 92.70 | 84.40 | 99.30 |
| **GeoLDM-StP** | **99.08±0.05** | **90.70±0.22** | **95.41±0.16** | **93.49±0.16** | **86.78** | **99.37** |

Baseline results are taken from original works, some only have one decimal places available.

We first generate molecules unconditionally with trained models. The results are presented in Table 1. For EDM, StP enhances the earliest EDM to surpass GeoLDM in generating valid molecules and outperforms the most recent RADM on QM9 across all metrics. For RADM, StP improves the quality of generated molecules with higher atom stability, molecular stability, and validity; the uniqueness is lower, potentially because the transformer backbone of RADM fits the training data strongly. For GeoLDM, StP improves performance across all metrics; in fact, **GeoLDM-StP outperforms the complicatedly designed SOTA flow-based baseline, GeoBFN, and achieves the overall best results**[3]**, establishing a new SOTA performance for 3D molecular diffusion.** It is worth noting that GeoLDM achieves higher atom stability than the dataset distribution on both QM9 and GEOM-Drugs, similar to GeoBFN on QM9. This may be because most atoms in the dataset are stable, and the model implicitly denoises toward the mode of the distribution (i.e., stable atoms). This is desirable in stable molecular generation. From a generative modeling perspective, a good model should closely match the data distribution. In this regard, GeoLDM-StP is still the closest to the dataset distribution compared to other methods (the difference for GeoLDM-StP is the lowest; $0.08\%$ on QM9 and $0.28\%$ on GEOM-Drugs). In addition, we provide the same per-size visualization as in Fig. 2 but with StP in Fig. 11 in Appendix A.3. Clearly, StP improves the quality of generated molecules across all sizes and significantly reduces the size-induced inconsistency. **In conclusion, StP consistently enhances the quality of generated molecules by improving both stability and validity across backbone diffusion models and benchmark datasets, demonstrat-**

---

[3]No single model achieves the best results across all columns; however, GeoLDM-StP demonstrates the best overall performance.

Table 2: Results on conditional generation tasks. The reported metric is MAE (lower is better). The overall best result is highlighted in **red**. Clearly, StP enhances the ability to generate molecules with desired properties and achieves the best performance for all 6 tasks.

| Property | $\alpha$ | $\Delta\varepsilon$ | $\varepsilon_{\text{HOMO}}$ | $\varepsilon_{\text{LUMO}}$ | $\mu$ | $C_v$ |
| Unit | Bohr$^3$ | meV | meV | meV | D | $\frac{\text{cal}}{\text{mol}}$K |
| --- | --- | --- | --- | --- | --- | --- |
| QM9 (Lower Bound) | 0.10 | 64 | 39 | 36 | 0.043 | 0.040 |
| EDM | 2.76 | 655 | 356 | 584 | 1.111 | 1.101 |
| EquiFM | 2.41 | 591 | 337 | 530 | 1.106 | 1.033 |
| GeoLDM | 2.37 | 587 | 340 | 522 | 1.108 | 1.025 |
| GeoBFN | 2.34 | 577 | 328 | 516 | 0.998 | 0.949 |
| RADM | 1.98 | 458 | 290 | 383 | 0.814 | 0.869 |
| RADM-StP | **1.96** | **425** | **282** | **360** | **0.792** | **0.861** |

**ing its effectiveness and generalizability. These results and findings highlight the importance, effectiveness, and generalizability of StP.**

**Efficient Sampling.** The sampling process is unnecessarily slow in the original implementations of EDM, GeoLDM, and RADM, especially for GEOM-Drugs. We provide an efficient implementation that accelerates the sampling process by at most $12.21\times$ (from 12.94 seconds per sample to 1.06 seconds). The details and numerical results are provided in Appendix D.1. We do not claim any novel technical contribution; it is provided solely as a resource for the community.

## 4.3 CONDITIONAL GENERATION

Following prior work Hoogeboom et al. (2022); Xu et al. (2023), we assess the effectiveness of StP in conditional generation tasks in QM9, specifically generating molecules with target properties. We report the Mean Absolute Error (MAE) using the pretrained predictor; a smaller deviation from QM9 (Lower Bound) indicates better performance. More details are described in D.2. We need to train a separate diffusion model for each conditional generation task. Due to computational constraints, we only evaluate the best-performing diffusion model on conditional tasks, RADM, with StP. The results are presented in Table 2. **Clearly, StP not only generates more valid and stable molecules, but also enhances the ability to generate molecules with desired property values; StP improves the performance of RADM on all six conditional generation tasks and establishes a new SOTA in conditional generation performance.**

## 4.4 ABLATION STUDY: COMPARISON WITH DIRECT NORMALIZATION

As discussed in Sec. 3.3, directly normalizing molecules using different normalization constants across molecular sizes distorts important structural information. To demonstrate its impact, we conduct an ablation study comparing the proposed StP (-StP) with direct data normalization on the QM9 dataset (-N). In particular, the atomic coordinates are normalized by $\gamma_N$ as discussed in Sec. 3.3. The results are presented in Table 3. For EDM, the results may at first seem counterintuitive: direct normalization leads to lower atom stability but higher molecule stability and overall validity compared with the original model. This occurs because ① there are more unstable atoms in larger molecules due to distortions in structural information(e.g., bond length), and no matter how many unstable atoms there are, it is just one unstable

Table 3: Results on comparison with direct normalization. The best result among the original, direct normalization, and StP is highlighted in **orange**. Clearly, StP achieves the best performance.

| | Atom Stab (%) | Mol. Stab (%) | Valid (%) |
| --- | --- | --- | --- |
| EDM | 98.7 | 82.0 | 91.9 |
| EDM-N | 98.6 | 85.8 | 93.1 |
| EDM-StP | **98.8** | **88.1** | **94.4** |
| RADM | 98.5 | 87.3 | 94.1 |
| RADM-N | 98.4 | 86.2 | 93.6 |
| RADM-StP | **98.6** | **87.6** | **94.2** |
| GeoLDM | 98.9 | 89.4 | 93.8 |
| GeoLDM-N | 98.9 | 90.1 | 94.3 |
| GeoLDM-StP | **99.1** | **90.7** | **95.4** |

molecule, and ② the benefits of harmonizing generative dynamics outweigh the harms of structural distortion for smaller molecules. For GeoLDM, similarly to EDM, we observe the same atom stability but higher molecule stability and validity. For RADM, we observe even worse performance with direct normalization; this is potentially due to the transformer architecture, causing the harms of structural distortion outweigh the benefits of reducing size-inconsistencies. **Overall, for all three diffusion models, StP consistently demonstrates the best performance. In conclusion, these re-**

**sults clearly illustrate the importance of StP, which not only harmonizes generative dynamics across different molecular sizes but also preserves essential chemical structural information.**

## 5 CONCLUSION

In this work, we identified a fundamental yet previously overlooked issue in diffusion-based 3D molecular generation: the inconsistency of denoising dynamics across molecular sizes. By decomposing the generative process into shape and atom type components, we revealed how molecules of different sizes establish structural targets at different rates, leading to invalid or unstable outputs. To address this, we proposed StP, Scaling the Prior, which rescales the noise distribution relative to molecular sizes. Unlike direct data normalization, StP preserves chemically meaningful information such as bond lengths while harmonizing denoising trajectories across molecular sizes. Our analysis and experiments demonstrate that StP significantly improves the validity, stability, and overall quality of generated molecules, without requiring additional architectural complexity.

**Limitations and Future Work.** While our work addresses size-induced inconsistency in diffusion-based 3D molecular generation, our work can be extended to flow-based 3D molecular generation. Flow-based 3D molecular generation usually uses the standard Gaussian distribution as the prior distribution for coordinates as well (Song et al., 2023; 2024; Dunn & Koes, 2024). Practically, it can work the same way that we use different variance levels for the prior Gaussian. Furthermore, applying to flow models, StP theoretically could be justified from the perspective of optimal transport, where StP can reduce the cost of transporting noise distributions across molecules of different sizes. In addition, the normalization factors used in Sec. 3.3 are not derived from physical or chemical principles. Exploring normalization factors with physical justification or chemical meaning could be an interesting direction for future work.

## ETHICS STATEMENT

This research is centered on the development and evaluation of diffusion models for 3D molecular generation. The study does not involve human participants, personal data, or any sensitive information that could raise concerns regarding privacy, security, or fairness. Moreover, no conflicts of interest, issues of legal compliance, or potentially harmful applications have been identified in connection with this work.

## REPRODUCIBILITY

The datasets used in this work are open-source. Upon acceptance of the paper, we will release the source code together with detailed instructions for dataset preparation, pre-trained models, and configuration files necessary to reproduce the main experiments. Along with the code, we will provide the values of scales, the values of $\gamma_N$ (see Sec. 3.3), and evaluation results for different molecular sizes. Comprehensive guidelines, including command-line examples for training and evaluation, will also be made available. All theoretical claims in this work are rigorously supported by proofs presented in the appendix.

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

## LLM Usage

LLMs have been used to assist with polishing the writing (e.g., grammar, typos, readability) and to help with writing code when given clear and detailed instructions. All LLM-generated texts and codes are verified before use by the authors (humans). All conceptual innovations, methodological designs, and research contributions are developed solely by the authors.

## A    Size-induced Inconsistencies

### A.1    Generation Quality Across Different Sizes

**Experimental Details.** We access the difference in generation quality by sampling $3,000$ molecules of each size with 3 SOTA diffusion models for 3D molecular generation, namely EDM, RADM, and GeoLDM. We follow the exact same setups and directly use the pretrained weights provided by the respective works.

**Additional Results.** We present size-induced inconsistencies in the molecule stability of generated molecules on QM9 with RADM and GeoLDM in Fig. 4 and Fig. 5, respectively, size-induced inconsistencies in the validity of generated molecules on QM9 for all 3 diffusion models in Fig. 6, Fig. 7, and Fig. 8, respectively, and size-induced inconsistencies in atom stability on GEOM-Drugs for all 3 diffusion models in Fig. 9. **It is clear that size-induced inconsistencies are present in various diffusion pipelines for 3D molecular generation.** Note that we do not present validity results for the GEOM-Drugs dataset because the validity is already very close to 1 across all molecule sizes. This aligns with the observation in Hoogeboom et al. (2022) that validity remains near 1 when evaluated over all possible sizes.

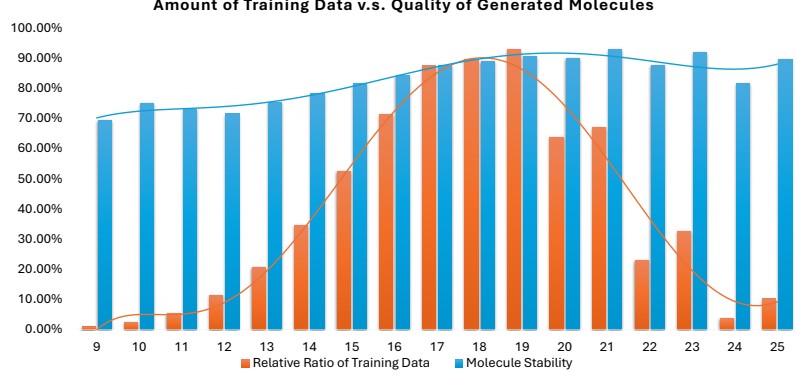

Figure 4: Molecule stability of generated molecules ($y$-axis) versus molecular size ($x$-axis) with RADM on QM9.

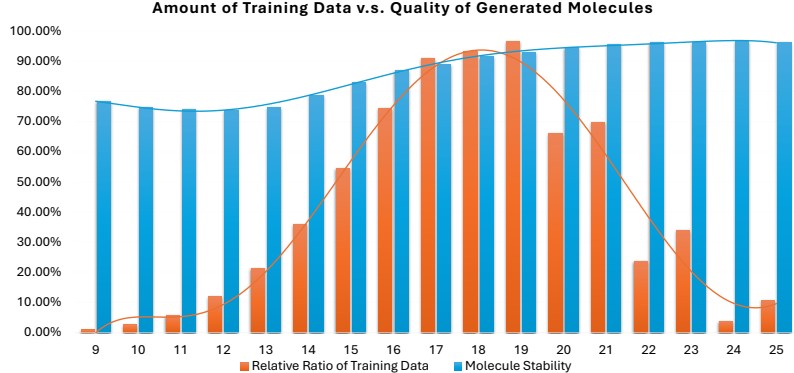

Figure 5: Molecule stability of generated molecules ($y$-axis) versus molecular size ($x$-axis) with GeoLDM on QM9.

### A.2    Generation Target Alignment Across Different Sizes

**Experimental Details.** We sample $1,000$ generation trajectories for each size with 3 SOTA diffusion models for 3D molecular generation, namely EDM, RADM, and GeoLDM. For latent diffusion

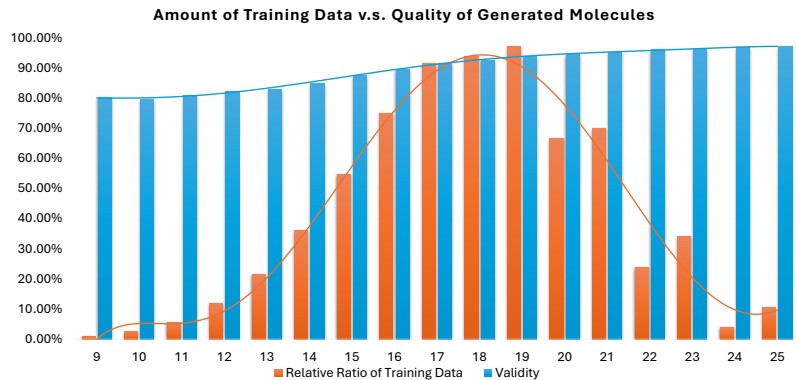

Figure 6: Validity of generated molecules ($y$-axis) versus molecular size ($x$-axis) with EDM on QM9.

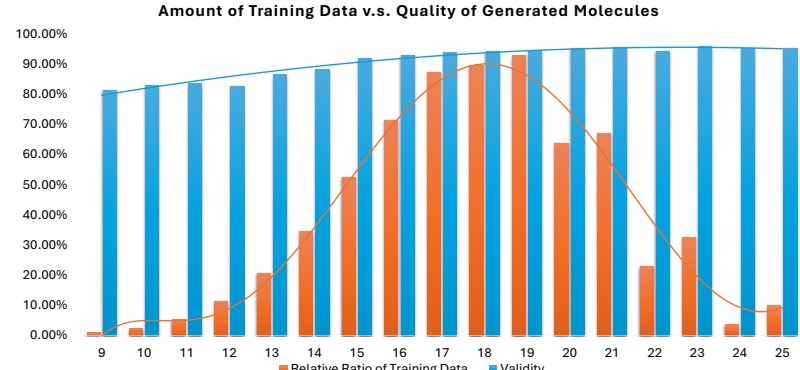

Figure 7: Validity of generated molecules ($y$-axis) versus molecular size ($x$-axis) with RADM on QM9.

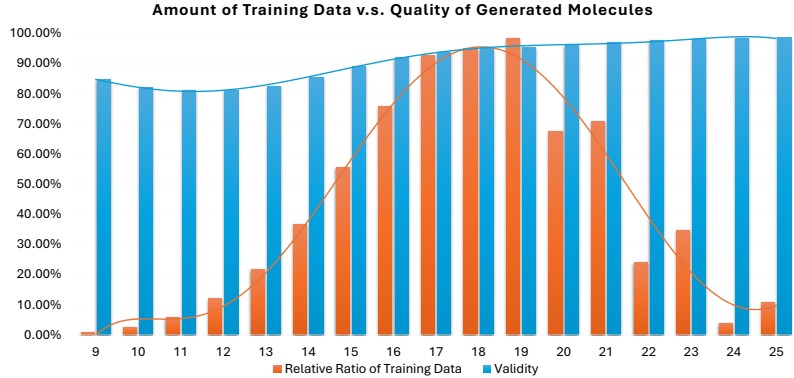

Figure 8: Validity of generated molecules ($y$-axis) versus molecular size ($x$-axis) with GeoLDM on QM9.

models (RADM and GeoLDM), the coordinates are represented directly in the 3D Euclidean latent space. For atom types, we obtain the concrete atom types by passing their latent representations through the decoder, which are then used to compute the alignment ratio. Note that tracking the full trajectory is memory intensive, so only 100 trajectories are sampled each time, and we average over 10 runs. It is observed that the alignment convergence patterns remain consistent across different runs. We follow the exact same setups and directly use the pretrained weights provided by the respective works.

**Additional Results.** We present size-induced inconsistencies in the generation target alignment for EDM in Fig. 12, for RADM in Fig. 13, and for GeoLDM in Fig. 14. Note that these figures are placed at the end of the paper due to space constraints. **It is clear that there are size-induced inconsistencies for all three SOTA diffusion models.** The inconsistencies in atom type alignment

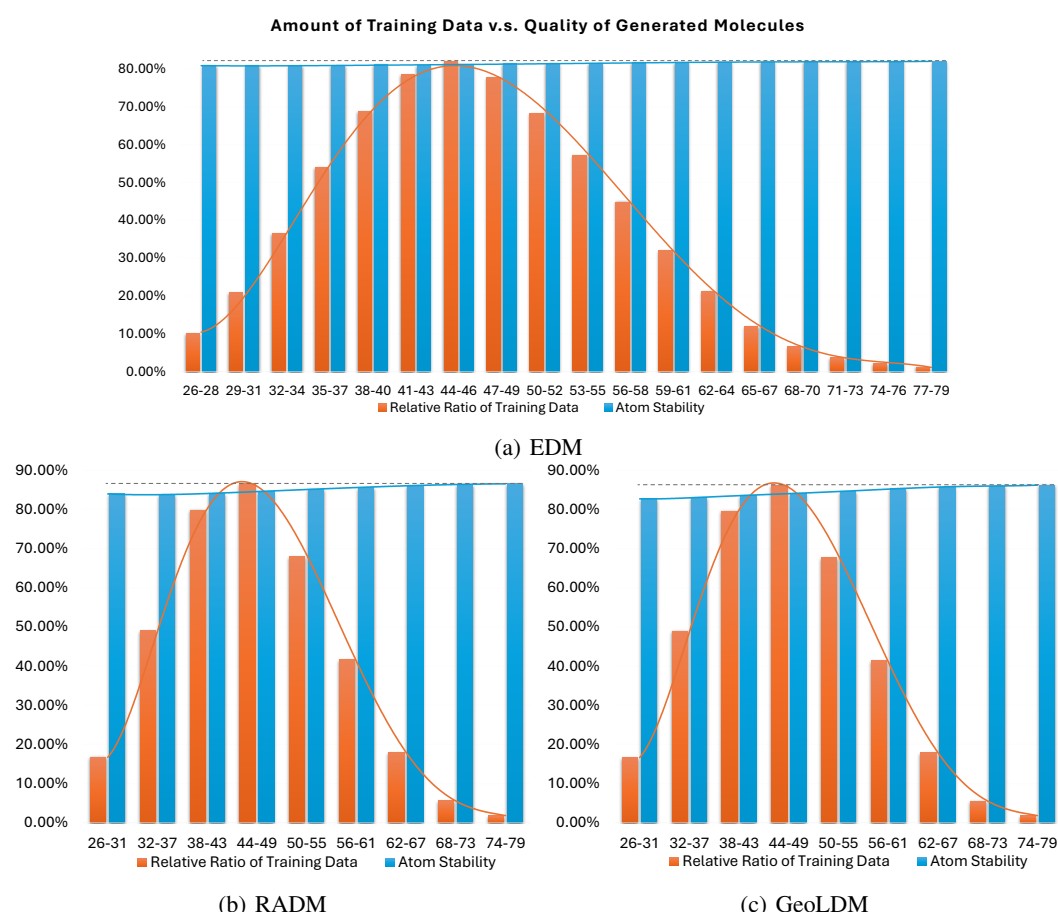

(a) EDM

(b) RADM

(c) GeoLDM

Figure 9: Atom stability of generated molecules ($y$-axis) versus molecular size ($x$-axis) with all 3 SOTA diffusion models on GEOM. For visualization, molecular sizes are grouped into inclusive bins. Note that the reported metric for sampling quality is now atom stability. While the numerical differences appear smaller compared to molecular stability, they are nonetheless pronounced and important.

observed on the GEOM-Drugs dataset for GeoLDM are less pronounced than that in EDM and RADM, because atom types in latent models are determined much slower, after the shapes have already stabilized; this is potentially because the autoencoder is untrained and the latent space is highly compact, as indicated by the authors in their GitHub repository with the provided pretrained weights. However, we can still observe the inconsistencies, especially by comparing the values of small sizes (e.g. 10 and 12) with that of large sizes (e.g. 45 and larger).

## A.3 STP: A LEARNING RESPECTIVE

For notational simplicity, we consider only the 3D coordinates so that $x \in \mathbb{R}^{3 \times N}$. In Sec. 2.1, the denoiser network learns to map the noisy sample to the noise added; mathematically, $\phi : (z_t, t) \mapsto \epsilon$, where $z_t^x = \alpha_t x + \sigma_t \epsilon$. However, this denoiser must implicitly learn a size-dependent scaling to account for the variation in molecular spatial scales. For example, at $t = 1$, starting from pure Gaussian noise, the denoiser needs to learn to contract coordinates for smaller molecules and expand them for larger ones, as illustrated in Fig. 10. Depending on the normalization of the data, this may also manifest as expanding more or contracting less for larger molecules. This inconsistency in the learning dynamics leads to a size-induced inconsistency in the generation quality as demonstrated in Fig. 2 in the main paper. StP mitigates this issue by scaling the prior Gaussian distribution so that the generative process learns consistent expansion or contraction across molecular sizes.

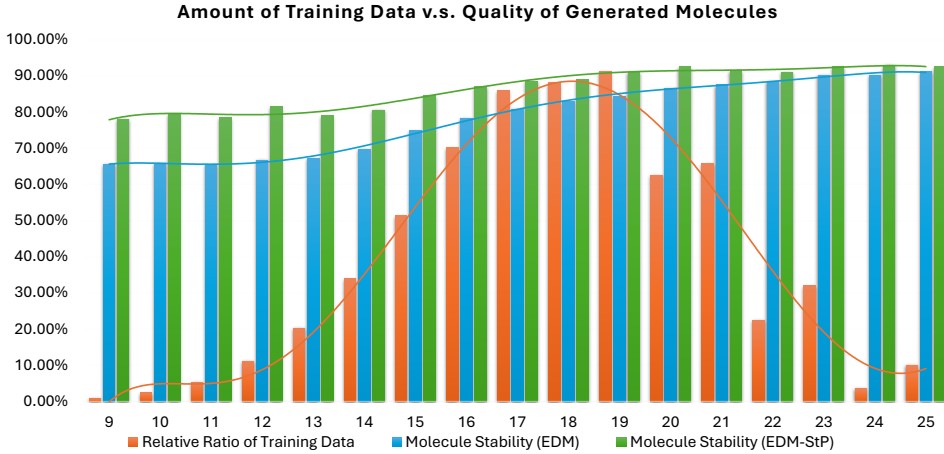

Figure 10: Illustration of different spatial scales between a molecule of size 91 and a molecule of size 5, compared to their respective Gaussian noise of the same size. The generative process and the denoiser network are trained to map pure Gaussian noise to realistic and valid molecular structures. However, they must learn a size-dependent behavior, contracting coordinates for smaller molecules and expanding them for larger ones. This requirement introduces inconsistency in the learning process, leading to size-induced inconsistencies in generation quality. StP mitigates this issue by scaling the spatial extent of the Gaussian prior for different sizes so that the generative model learns the same dynamics consistently across molecular sizes.

In Fig. 11, we provide the same per-size visualization as in Fig. 2 of the main paper, but with StP, to illustrate its effectiveness in reducing size-induced inconsistency. As shown in Fig. 11, the generation quality with StP remains roughly the same for molecular sizes greater than 17, whereas it continues to increase without StP. In addition, StP significantly improves the generation quality for smaller molecules. Larger molecules still exhibit better performance under the same data scarcity, potentially because the backbone neural networks generalize more effectively for larger molecules due to the increased number of interatomic interactions.

**Amount of Training Data v.s. Quality of Generated Molecules**

Figure 11: Molecule stability of generated molecules ($y$-axis) versus molecular size ($x$-axis) with EDM and EDM-StP on the QM9 dataset (Ramakrishnan et al., 2014). Clearly, StP mitigates the size-induced inconsistency in generation quality. The generation quality with StP remains roughly the same for molecular sizes greater than 17, whereas it continues to increase without StP. In addition, StP significantly improves the generation quality for smaller molecules.

## A.4 CONVEX HULL

Given a set of 3D coordinates $x = (x_1, x_2, \ldots, x_N) \in \mathbb{R}^{N \times 3}$, the *convex hull* is the smallest convex polytope that contains all points $x_i$. Formally, the convex hull is defined as

$$\text{Hull}\left(\{x_1, \ldots, x_N\}\right) = \left\{\sum_{i=1}^{N} \lambda_i x_i \mid \lambda_i \geq 0, \sum_{i=1}^{N} \lambda_i = 1\right\}. \tag{13}$$

918 A key property is that the convex hull can always be described by a subset of the original points,
919 called vertices or extreme points. These are the points that cannot be written as convex combina-
920 tions of the others. **Geometrically, they lie on the "outer surface" of the configuration. All**
921 **other points in the set $\{x_i\}$ are contained strictly inside the convex hull** and can be expressed as
922 convex combinations of the vertices. The readers are refereed to De Berg et al. (2008) for fur-
923 ther details on convex hulls and the algorithms to compute convex hull vertices. In this work,
924 to find convex hull vertices, we use the algorithm implemented in the *scipy* Python library with
925 *scipy.spatial.ConvexHull()*. Specifically, it performs the QHull (divide and conquer) algorithm.

## B DISCUSSION ON LATENT DIFFUSION MODELS

931 Latent diffusion models encode molecular coordinates and atom features $(\boldsymbol{x}, \boldsymbol{h})$ into a lower-
932 dimensional latent space. Existing approaches, such as GeoLDM and RADM, still treat coordi-
933 nates and atom features separately, keeping coordinates in the 3D Euclidean space within the latent
934 representation as $\boldsymbol{x}' \in \mathbb{R}^{(N-1)\times 3}, \boldsymbol{h}' \in \mathbb{R}^{N \times d'}$, where typically $d' < d$.

935 GeoLDM aims to respect the rotational symmetries inherent in molecular data (translation symmetry
936 can always easily be ensured by zero-mean). Specifically, GeoLDM employs an equivariant encoder
937 $\mathcal{E}(\cdot)$ and decoder $\mathcal{D}(\cdot)$ such that

$$\mathcal{E}(R \cdot \boldsymbol{x}) = R \cdot \mathcal{E}(\boldsymbol{x}) \quad \text{and} \quad \mathcal{D}(R \cdot \boldsymbol{x}) = R \cdot \mathcal{D}(\boldsymbol{x}), \tag{14}$$

942 where $R$ is a representation of the $SO(3)$ group. For simplicity, we ignore translations since they
943 can be easily handled by centering coordinates to zero mean, and we omit atomic features since they
944 are scalar values that remain unchanged under Euclidean transformations. In essence, equivariance
945 guarantees that when the molecular point cloud is rotated, the latent representation rotates in the
946 same way. To achieve this, the latent space must carry a valid representation of $SO(3)$. The small-
947 est nontrivial irreducible representation (irrep) of $SO(3)$ is the 3D vector representation. **Conse-**
948 **quently, maintaining rotational equivariance requires the latent representation of coordinates**
949 **to remain in 3D coordinate form.**

950 On the other hand, RADM aims to standardize molecular configurations into a fixed orientation in
951 an unsupervised manner. After standardization, the 3D coordinates can be encoded into lower di-
952 mensional spaces. However, RADM also still remains in 3D coordinate form. Despite consideration
953 of the rotation symmetry of the molecular data, the final generation goal is to generate molecules
954 with their full 3D atomic configurations. Molecular configurations are very sensitive: small errors in
955 3D coordinates can lead to totally invalid or unstable molecules. If 3D coordinates were compressed
956 into a lower-dimensional latent space (e.g., one or two dimensions), this would inevitably result in
957 loss of geometric information. Since such information cannot be perfectly recovered when map-
958 ping back to 3D space, the generated structures would risk being distorted or chemically unrealistic.
959 **Therefore, to minimize the reconstruction error in atomic coordinates, it is desirable to have**
960 **the coordinates still represented in 3D in the latent space.**

960 Due to similar reasons, it is also undesirable for the latent space to rely on overly complicated trans-
961 formations of the coordinates. If the encoding step introduces highly nonlinear or abstract latent
962 representations of molecular geometry, it may obscure or distort the essential spatial relationships
963 between atoms. Since the generative model must ultimately recover precise 3D configurations, such
964 complexity can make the reconstruction inaccurate. While small inaccuracies may be tolerable in
965 other application domains, such as image generation, they are intolerable for molecular generation.
966 **Therefore, it is most effective to maintain a latent representation that stays close to the orig-**
967 **inal 3D coordinate configuration. In fact, in both GeoLDM and RADM, it can be seen that,**
968 **while the atomic features are encoded into lower-dimensions, the coordinates rarely change**
969 **in the latent space.** Quantitatively, we can measure the relative $l_2$ difference between the original
970 coordinates and their latent representations: $\frac{||\boldsymbol{x} - \mathcal{E}(\boldsymbol{x})||_2}{||\boldsymbol{x}||_2}$. We report the average relative $l_2$ difference
971 over the entire training dataset in Table 4. Clearly, the encoders are almost the identity function for
coordinates; the atomic coordinates rarely change in latent space.

Table 4: Relative $L_2$ difference between the original atomic coordinates and the latent coordinates. For RADM, we also apply the rotater to the original coordinates to ensure rotational alignment, so that the error reflects structural discrepancies rather than arbitrary orientation differences.

| Auto-Encoder | QM9 | GEOM-Drugs |
|---|---|---|
| RADM | 2.87% | 0.03% |
| GeoLDM | 1.95% | 1.23% |

## C  PROOF TO PROPOSITION

**Proposition 3.2.** *The marginal distributions at any time $t$ of Scaling the Prior are equivalent to those of normalizing the data $\boldsymbol{x}$ by a factor of $\frac{1}{\gamma_N}$ before the forward process, and subsequently unnormalizing after sampling.*

*Proof.* Mathematically, it is equivalent to proving the following:

$$q_\gamma(\boldsymbol{z}_t^x \mid \boldsymbol{x}) = q\left(\frac{1}{\gamma_N}\boldsymbol{z}_t^x \;\middle|\; \frac{1}{\gamma_N}\boldsymbol{x}\right). \tag{15}$$

Note that:

$$
\begin{aligned}
q_\gamma(\boldsymbol{z}_t^x \mid \boldsymbol{x}) &= \mathcal{N}\left(\boldsymbol{z}_t^x; \sqrt{\alpha_t}\boldsymbol{x}, \gamma_N^2\sigma_t^2\mathbf{I}\right) \\
&\propto \exp\left(-\frac{\left\|\boldsymbol{z}_t^x - \sqrt{\alpha_t}\boldsymbol{x}\right\|^2}{2\gamma_N^2\sigma_t^2}\right) \\
&= \exp\left(-\frac{1/\gamma_N^2\left\|\boldsymbol{z}_t^x - \sqrt{\alpha_t}\boldsymbol{x}\right\|^2}{2\sigma_t^2}\right) \\
&= \exp\left(-\frac{\left\|\frac{1}{\gamma_N}\boldsymbol{z}_t^x - \frac{1}{\gamma_N}\sqrt{\alpha_t}\boldsymbol{x}\right\|^2}{2\sigma_t^2}\right) \\
&\propto \mathcal{N}\left(\frac{1}{\gamma_N}\boldsymbol{z}_t^x; \frac{\sqrt{\alpha_t}}{\gamma_N}\boldsymbol{x}, \sigma_t^2\mathbf{I}\right) \\
&= q\left(\frac{1}{\gamma_N}\boldsymbol{z}_t^x \;\middle|\; \frac{1}{\gamma_N}\boldsymbol{x}\right).
\end{aligned}
\tag{16}
$$

$\square$

## D  EXPERIMENTAL DETAILS AND ADDITIONAL RESULTS

### D.1  EFFICIENT SAMPLING

In the implementation of sampling in EDM, as as well as in RADM and GeoLDM which inherit from EDM, the sampling process becomes unnecessarily slow when the maximum molecular size is much larger than that of most molecules in the dataset (e.g. the average molecular size in GEOM-Drugs is 44.4 while the maximum is 181). This is unnecessarily slow because during sampling, the method constructs graphs of the dataset's maximum size $N_{\max}$, padding with dummy nodes for parallelization as described in the pseudo code in Algorithm 1. **However, within a given batch, the actual maximum size $N_{\text{batch\_max}}$ is often much smaller than $N_{\max}$. Noticing this bottleneck, we can accelerate the sampling process by constructing graphs only up to the batch maximum size $N_{\text{batch\_max}}$. More importantly, when sampling a large number of molecules, we can first sample molecular sizes and then sort them so that each batch contains molecules with similar sizes, thereby avoiding unnecessary computation.** If array storage and future parallel process are needed, the generated results can be padded with dummy nodes to be the maximum size from the overall sampled molecular sizes or the dataset max $N_{\max}$ for simplicity. The pseudo code for accelerated sampling is presented in Algorithm 2. Notice that this sampling is mathematically equivalent to the original sampling in the implementation in EDM; it is just an efficient implementation.

Table 5: Generation time per sample for original vs. efficient methods.

| Model | Original (s) | Efficient (s) | Speedup ($\times$) |
|---|---|---|---|
| EDM | 12.94 | 1.06 | 12.21 |
| RADM | 2.11 | 0.59 | 3.58 |
| GeoLDM | 13.03 | 1.07 | 12.18 |

We follow the standard sampling practice used in all three SOTA diffusion models. Specifically, we generate $10,000$ molecules with a batch size of $100$ on GEOM-Drugs. Noticeably, this accelerate the generation by $12.21\times$ for EDM, $3.58\times$, for RADM, almost $12.18\times$ for GeoLDM. We provide the statistics in Table 5. The timing results are recorded on a single NVIDIA RTX A6000 48GB Graphics Card. **This acceleration is crucial, as it enables size-dependent studies in this work that would otherwise be computationally prohibitive.**

**Training.** Note that during training, the graphs are constructed based on the batch maximum molecular size; thus, it is already efficient. Although it is possible to not randomly shuffle the training set and sort it, this may introduce systematic bias in the training process. Moreover, in most cases, the batch maximum is still relatively small and all molecular sizes within the batch are close, so the additional computational overhead is tolerable.

---

**Algorithm 1** Original EDM Sampling Implementation

---

**Require:** $M$ (total sample size), $M_{\text{batch}}$ (batch size), $P_{\text{size}}$ (distribution of molecule sizes), $N_{\max}$ (maximum molecular size), $\psi$ (trained diffusion model with sampling method)

1: Initialize all_samples $\leftarrow [\ ]$
2: num_batches $\leftarrow M/M_{\text{batch}}$          ▶ *Assume $M_{batch}$ divides $M$*
3: **for** $i = 1$ to num_batches **do**
4:     $\mathbf{N}_{\text{batch}} \in \mathbb{Z}^{M_{\text{batch}}} \leftarrow \text{Sample}(P_{\text{size}}, M_{\text{batch}})$      ▶ *Sample $M_{batch}$ times from $P_{size}$*
5:     Construct $M_{\text{batch}}$ initial graphs $\mathcal{G}$, all of size $N_{\max}$ with node masks based on $\mathbf{N}_{\text{batch}}$
6:     batch_samples $\leftarrow \psi.\text{sample}(\mathcal{G})$
7:     all_samples $\leftarrow$ all_samples $\cup$ batch_samples      ▶ *Append batch samples*
8: **end for**
9: **return** all_samples

---

**Algorithm 2** Efficient Sampling Implementation

---

**Require:** $M$ (total sample size), $M_{\text{batch}}$ (batch size), $P_{\text{size}}$ (distribution of molecule sizes), $N_{\max}$ (maximum molecular size), $\psi$ (trained diffusion model with sampling method)

1: Initialize all_samples $\leftarrow [\ ]$
2: $\mathbf{N} \in \mathbb{Z}^M \leftarrow \text{Sort}(\text{Sample}(P_{\text{size}}, M))$      ▶ *Sample $M$ times from $P_{size}$ and sort*
3: num_batches $\leftarrow M/M_{\text{batch}}$      ▶ *Assume $M_{batch}$ divides $M$*
4: **for** $i = 1$ to num_batches **do**
5:     $\mathbf{N}_{\text{batch}} \leftarrow \mathbf{N}[(i-1) \cdot M_{\text{batch}} + 1 : i \cdot M_{\text{batch}}]$      ▶ *Get a batch of molecular sizes by indexing*
6:     $N_{\text{batch\_max}} = \mathbf{N}_{\text{batch}} \leftarrow \mathbf{N}[i \cdot M_{\text{batch}}]$      ▶ *Maximum molecular size in the current batch*
7:     Construct $M_{\text{batch}}$ initial graphs $\mathcal{G}$, all of size $N_{\text{batch\_max}}$ with node masks based on $\mathbf{N}_{\text{batch}}$
8:     batch_samples $\leftarrow \psi.\text{sample}(\mathcal{G})$
9:     batch_samples $\leftarrow \text{Pad}(\text{batch\_samples}, \mathbf{N}[M])$      ▶ *Pad generated samples to max size*
10:    all_samples $\leftarrow$ all_samples $\cup$ batch_samples      ▶ *Append batch samples*
11: **end for**
12: **return** all_samples

---

### D.2 CONDITIONAL GENERATION

Following prior work, EDM (Hoogeboom et al., 2022), the objective of conditional generation is to generate molecules with target properties. Mathematically, this conditional generation in EDM is to generate $\boldsymbol{x}, \boldsymbol{h} \sim p(\boldsymbol{x}, \boldsymbol{h} \mid c)$ given some desired property $c$. The optimization lower bound for the conditional case can be defined as $\log p(\boldsymbol{x}, \boldsymbol{h} \mid c) \geq \mathcal{L}_{c,0} + \mathcal{L}_{c, \text{base}} + \sum_{t=1}^{T} \mathcal{L}_{c,t}$. The main difference is that the denoiser $\hat{\boldsymbol{\epsilon}}_t = \phi(\boldsymbol{z}_t, [t, c])$ takes an additional input, a property $c$, which is concatenated to the node features. Given a trained conditional model, sampling is done by first sampling the

molecular size (number of atoms $M$) and a property value $c$ from a distribution $c, M \sim p(c, M)$ that is inferred from the training partition. Then, given $c$ and $M$, we can generate molecules using conditional models.

**Implementation Details.** We follow the exact experimental setup as in EDM; RADM and GeoLDM all follow the exact same setup as well. Specifically, the QM9 training set is evenly split: an EGNN is trained on the first half as a property predictor, while the generative model is trained on the second half. After training, the predictor is used to evaluate the molecules generated by the model. We use the exact same EGNN model and hyperparameters as in EDM to train the property predictor.

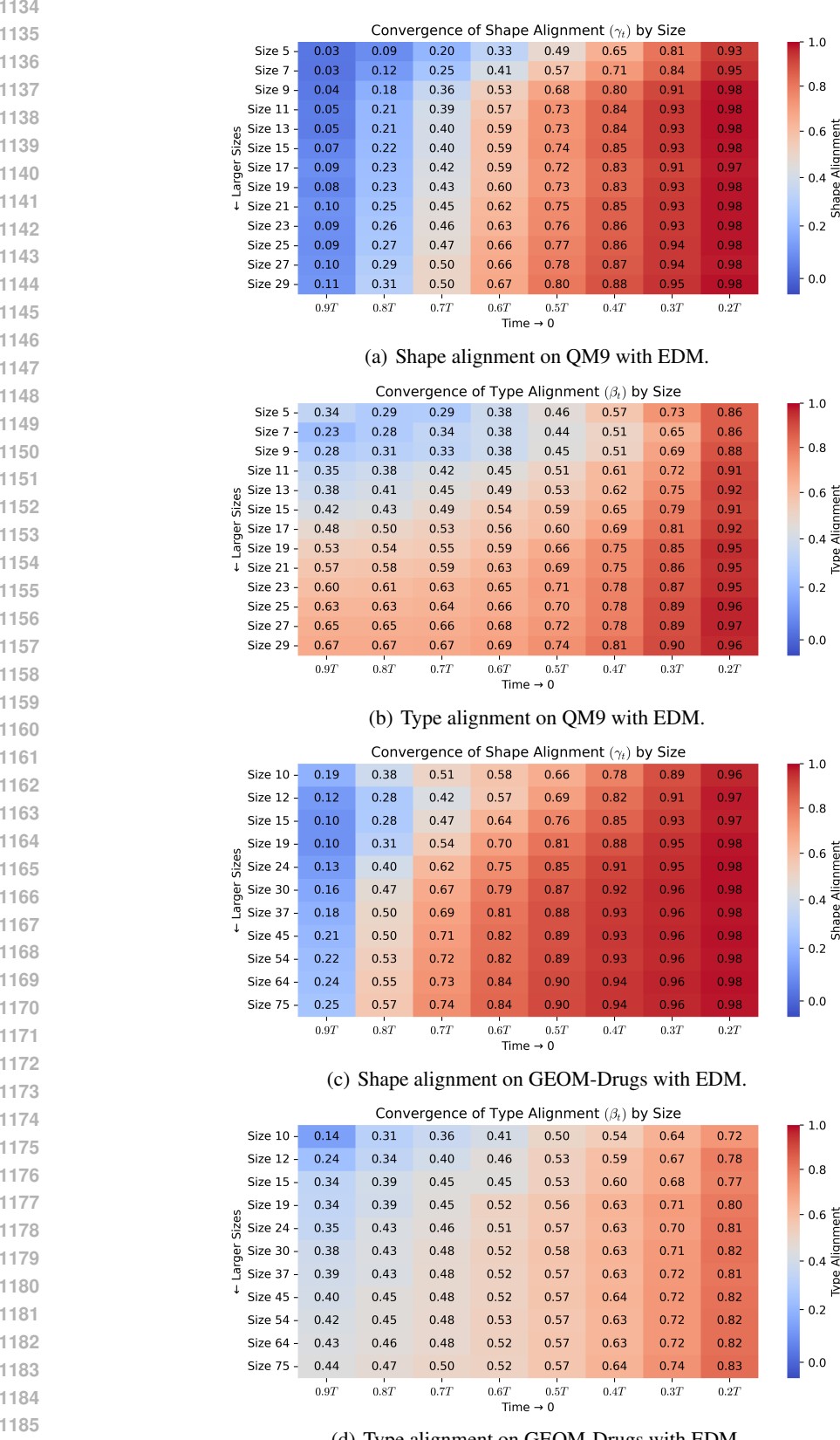

(a) Shape alignment on QM9 with EDM.

(b) Type alignment on QM9 with EDM.

(c) Shape alignment on GEOM-Drugs with EDM.

(d) Type alignment on GEOM-Drugs with EDM.

Figure 12: Convergence of generation target alignment with EDM.

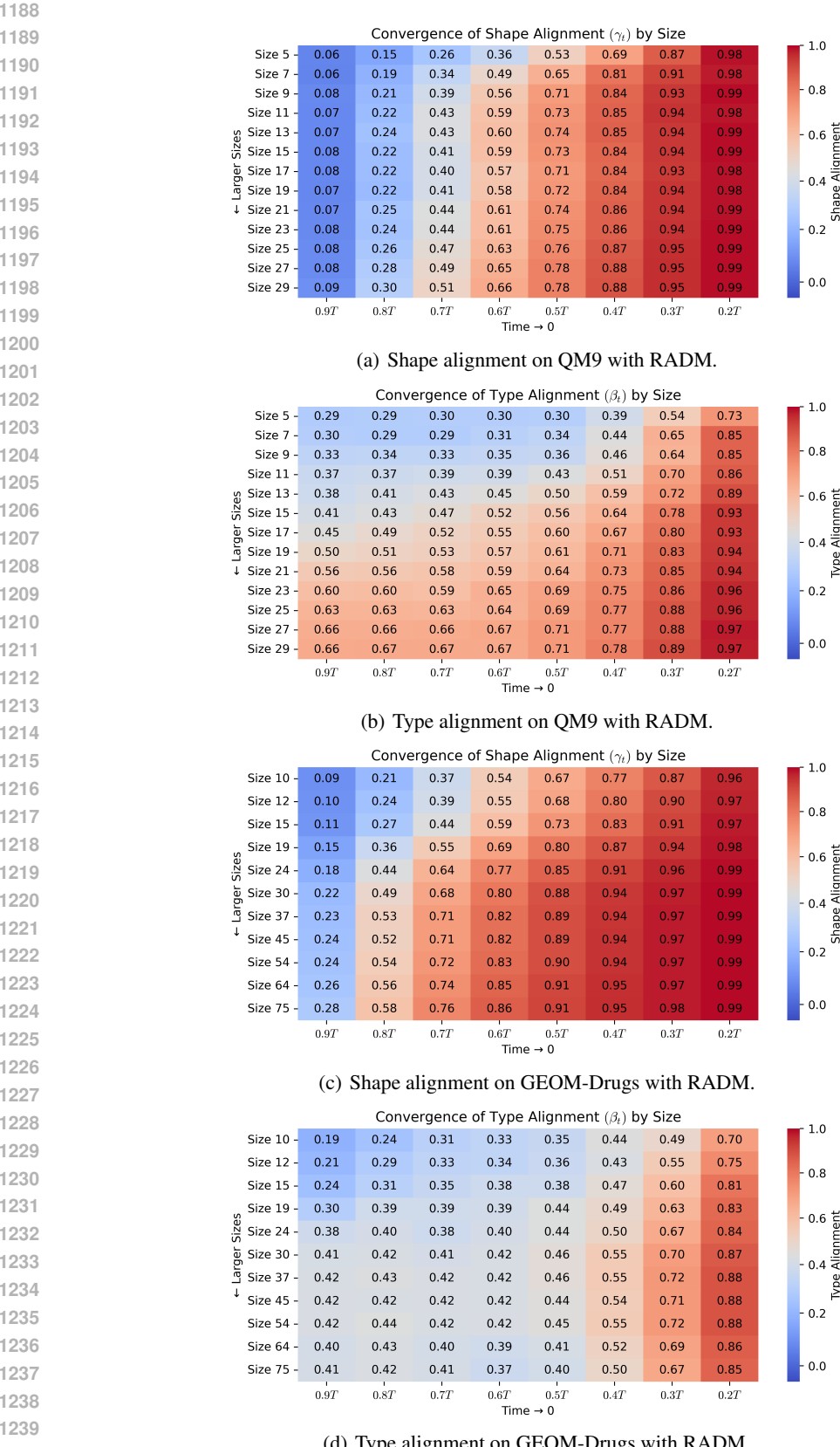

(a) Shape alignment on QM9 with RADM.

(b) Type alignment on QM9 with RADM.

(c) Shape alignment on GEOM-Drugs with RADM.

(d) Type alignment on GEOM-Drugs with RADM.

Figure 13: Convergence of generation target alignment with RADM.

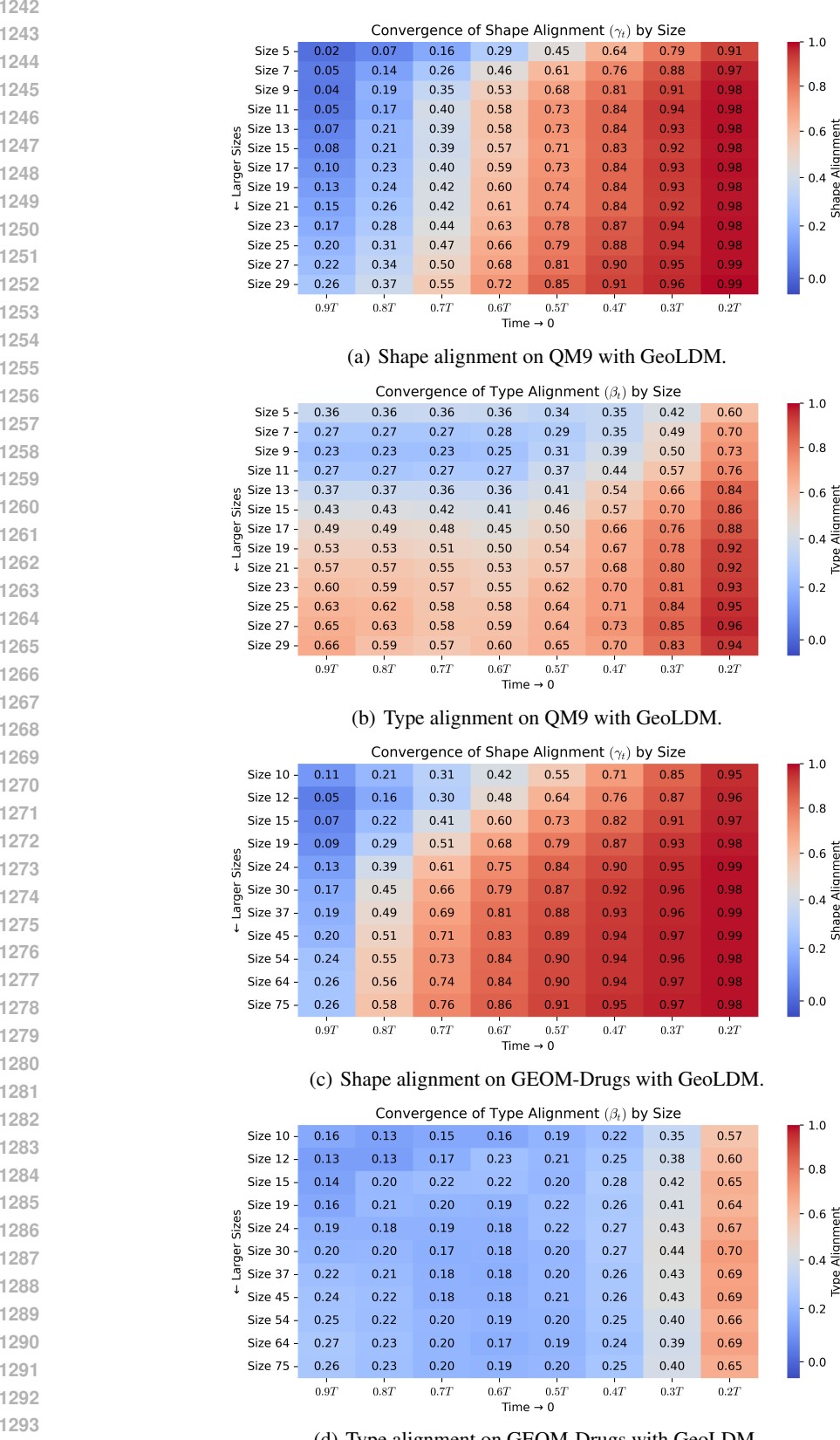

(a) Shape alignment on QM9 with GeoLDM.

(b) Type alignment on QM9 with GeoLDM.

(c) Shape alignment on GEOM-Drugs with GeoLDM.

(d) Type alignment on GEOM-Drugs with GeoLDM.

Figure 14: Convergence of generation target alignment with GeoLDM.

