# OpenReview forum: "Scaling the Prior: Size-Consistent Geometric Diffusion for 3D Molecular Generation"
_ICLR.cc/2026/Conference — Submitted to ICLR 2026_

### Official Review · Reviewer_5mF6 · 2025-10-24

**Soundness:** 2
**Presentation:** 2
**Contribution:** 2
**Rating:** 2
**Confidence:** 4

**Summary:**

This paper identifies a size-dependent inconsistency in diffusion-based molecular generation, where smaller molecules converge structurally slower than larger ones during both training and sampling. To address this, the authors propose StP, a simple modification that rescales the Gaussian prior variance based on molecular size, harmonizing the denoising trajectories across different molecular scales. This approach improves the consistency and quality of generated molecules from the experiments on QM9 and GEOM-Drugs, and could potentially extend to larger biomolecular systems such as proteins.

**Strengths:**

1. This paper provides interesting observations on the relation between molecule sizes and proximal structural convergence speed.
2. Experimental results show that StP works on some baseline diffusion models.

**Weaknesses:**

1. The paper’s notion of “convergence” is heuristic rather than principled. The proposed $\gamma_t$ and $\beta_t$ only measure correlations between predicted clean samples and final outputs, which depend on model training and not on intrinsic diffusion dynamics. Thus, the claim that smaller molecules “converge slower” lacks a rigorous statistical or theoretical foundation.
2. The use of convex hull–based scaling lacks physical justification and statistical robustness; it is an arbitrary geometric proxy for molecular size rather than a chemically meaningful measure.
3. While I appreicate authors' effort on observations and experiments, the reported gains are marginal on QM9 and GEOM-Drug generation.

**Questions:**

See Weakness

**Details Of Ethics Concerns:**

None.

---

> ### Author Response · Authors · 2025-11-14
> **Responses to Reviewer 5mF6**
>
> Thank you so much for your comments! We have revised the paper, and we provide our responses below.
>
> ### Weaknesses
> - W1: Notion of “convergence” is heuristic
>     - **We believe that the presented findings are statistically reliable and sufficiently representative, to demonstrate the inconsistency in the convergence of the generative process.** We have provided several results (Figures 3, 7, 8, and 9) across **different datasets and backbone models** to demonstrate that such size-inconsistent convergence is indeed present in 3D molecular diffusion. **For each molecular size, 1,000 trajectories were generated to produce these results.** All models are well-trained, and we directly use the pretrained weights provided by the original works. These measurements clearly reveal that there is a size inconsistency for the generative process.
>     - **We respectfully disagree with the comment that our proposed measurements depend solely on model training rather than on intrinsic diffusion dynamics.** If, by intrinsic diffusion dynamics, you refer to the theoretical diffusion dynamics, we believe this concept is not directly applicable to the problem investigated in our work. The theoretical formulations of diffusion models (e.g., SDE, ODE, Fokker–Planck equation, continuity equation, etc.) are defined on a fixed-dimensional metric space. Our work specifically aims to address this limitation and mitigate the challenges arising from varying input sizes (varying dimensions) in 3D molecular generation.
>
> - W2: The use of convex hull
>     - **Our experimental results in Table 1 show that the use of convex hull–based scaling is effective, improving both the stability and validity of all three diffusion backbone models.** In addition, our observations and proposed mitigation are independent of the specific scaling factor used. The convex hull discussion occupies only one paragraph in Section 3, which spans more than three pages.
>         - The main contribution of this work is to reveal the size-inconsistency problem in 3D molecular generation and to propose an effective remedy. **Convex hull-based scaling is a good geometric proxy as it represents the spatial extent of the 3D molecules.** Our work focuses on using deep generative models for 3D molecular generation; we humbly think that this convex hull-based scaling is natural and sufficient for our purpose. Researchers can follow up on our work and propose scaling factors with physical justification or chemical meaning. We have included this as future work in Sec. 5 in the revised paper (marked in red).
>
> - W3: Reported gains
>     - We appreciate your comments. **However, we do not find the gains marginal. It is not reasonable to evaluate the significance of the improvement solely based on the absolute numbers.** The magnitude of improvement must be interpreted in the context of the task.
>     - **It can be observed from Table 1 that many of our improvements are at least comparable to those reported in previous works, achieved using a simple method that does not incur additional computational costs. We therefore do not consider our improvement to be marginal and respectfully hope that you will reconsider your assessment.**
>         - For example, StP improves atom stability on GEOM-Drugs with GeoLDM by 2.38%, whereas GeoBFN (the previous SOTA) achieves an improvement of 1.30%. A recent diffusion-based work, RADM (ICML 2025) [1], reports an improvement of 0.6%.
>
> [1] Scalable Non-Equivariant 3D Molecule Generation via Rotational Alignment, ICML 2025

---

> > ### Author Response · Authors · 2025-11-25
> > **Follow-up on Our Reponses - Reviewer 5mF6**
> >
> > Dear Reviewer 5mF6,
> >
> > We are grateful for the time and attention you devoted to reviewing our work.
> >
> > - **It has already been 10 days since we posted our rebuttal, so we wanted to follow up and see if you have any remaining concerns.**
> >
> > - Specifically, we have taken the following steps to address your comments:
> >     - We have clarified that the notion of “convergence” is statistically significant and that we used pretrained, well-trained diffusion models from existing state-of-the-art works directly.
> >     - We have explained that the use of the convex hull is merely one particular design choice and only a small component of our overall framework. Additionally, it works very well with 3D molecular diffusion models.
> >     - We have shown that our reported performance gains are not marginal.
> >
> > - If there are no further concerns, we respectfully hope that you could reconsider your rating, as these weaknesses seem to arise from a misunderstanding.
> >
> > Best Regards
> >
> > Authors

---

> > > ### Comment · Reviewer_5mF6 · 2025-11-25
> > >
> > > Dear Authors,
> > >
> > > Thank you for your thorough response.
> > >
> > > I understand that the convergence defined in the network is calculated by well-pretrained diffusion models. However, since larger molecules have a larger spatial extent, adding standard Gaussian noise disrupts their global structure less than it does for smaller molecules. Consequently, for larger molecules, the model encounters inputs with a naturally higher signal-to-noise ratio (correlation with clean data) during training, which in a sense is exactly what this papers observers. Therefore, the observation that larger molecules stabilize earlier is an expected behavior of the noise schedule rather than a surprising discovery. While I appreciate the insight and it is intuitive, I hate to say that the technical novelty is a bit limited.
> > >
> > > Regarding the experimental results, while you argue that the improvements are significant, I remain concerned that the benchmarks used (QM9 and GEOM-Drugs) are already over-saturated. Marginal gains on these datasets make it difficult to accurately assess the true contribution of the proposed method. To convincingly demonstrate the effectiveness of the StP trick, validations on more challenging or less saturated datasets would be necessary.
> > >
> > > Still, I appreciate authors' response and nice writings in the paper, I raised my scores to 4. If, the authors could provide some initial results on larger scale datasets, such as protein dataset, I would consider raise my scores.
> > >
> > > Best,
> > > Reviewer 5mF6

---

> ### Author Response · Authors · 2025-11-25
> **Second Round Responses to Reviewer 5mF6**
>
> Thank you so much for your time and effort in reviewing our paper and for your willingness to discuss it.
>
> - Intuitive insight of StP and technical novelty
>     - This is subjective, and we completely understand if you feel this way. However, **we would like to emphasize that although the finding is intuitive, it has not been mentioned or addressed in any previous work. It may feel intuitive in part because we conducted the study, developed a way to quantify it, and presented the results nicely and clearly so that it is easy and clear to follow.** Being simple, insightful, and effective, in our view, constitutes the same level of novelty as being complex.
>
> - Saturated Results
>     - We fully understand your concern. Although there has been some pioneering work in this field, we believe our contribution remains insightful and pushes the boundary forward by a meaningful step.
>     - **As we pointed out previously, the improvement on the QM9 dataset is comparable to that of prior work.** We understand your view that the dataset may appear “saturated,” as the absolute improvements reported by all prior works and StP are relatively small (less than 0.4% for atom stability for all methods). **Nevertheless, on this dataset, we still observe clear improvements with StP (up to a 6.07% improvement in molecular stability on EDM with StP, and a 1.3% improvement on GeoLDM with StP.).**
>     - **If you refer to our results for the GEOM-Drugs dataset in Table 1, the improvement is substantial.** For example, **GeoLDM-StP improves atom stability from 84.40% to 86.78%**, which is substantial. In comparison, the improvement in atom stability on QM9 is typically less than 0.2% for all prior works compared to previous SOTA. In addition, under the same fair comparison setting, **the previous SOTA, GeoBFN, achieves an atom stability of 85.60% while the validity is only 92.08% (86.78% and 99.37%, respectively, for GeoLDM-StP).** GeoLDM-StP clearly shows a significant improvement and establishes a new SOTA performance despite being a simple architecture.
>     - **A more significant improvement on GEOM-Drugs further reinforces our claim and the effectiveness of StP.** The GEOM-Drugs dataset spans a wider range of molecule sizes (see Fig. 9 in Appendix A.1 in the revised paper for the size distribution), making size-related differences more pronounced than in the QM9 dataset. This further reinforces our claim that such size-induced inconsistency hinders performance, and our results in Table 1 demonstrate the effectiveness of StP.
>     - **Our work focuses on 3D molecular generation (following prior works, e.g. [1–5], and using exactly the same datasets), whereas protein generation differs substantially from our task.** Most existing de novo protein generation methods do not operate directly on raw coordinate space as we do in 3D molecular generation. Instead, they generate sequences that are later folded (e.g., [6]), design backbones that are subsequently sequenced (e.g., [7, 8]), use diffusion models on internal/angular representation of the backbone (e.g. [9]), or apply diffusion in a latent space without explicit coordinates in diffusion (e.g., [10]). While there may be a few all-atom protein generation works capable of generating raw coordinates (e.g., [11]), they are not as straightforward as in 3D molecular generation, where we simply generate all atom coordinates and node features. **Therefore, StP is not directly applicable.**
>     - **Given the results on GEOM-Drugs and the additional explanations we provide, we believe that the current experiments are convincing.**
>         - **We will also do our best to identify de novo all-atom protein generation works that are compatible with StP.** However, we emphasize that our work requires training diffusion models and that all-atom protein generation is significantly more computationally expensive than 3D molecular generation. **Given the time constraints of the rebuttal period and our limited computational resources, we must be frank that it is very difficult for us to produce new results on 3D protein generation, which is a fundamentally different task from 3D molecular generation.**

---

> > ### Author Response · Authors · 2025-11-25
> > **Second Round Responses to Reviewer 5mF6 - References**
> >
> > [1] EDM: E(3) Equivariant Diffusion Model for Molecule Generation in 3D, ICML 2022
> >
> > [2] Geometric Latent Diffusion Models for 3D Molecule Generation, ICML 2023
> >
> > [3] Equivariant Flow Matching with Hybrid Probability Transport, NeurIPS 2023
> >
> > [4] Unified Generative Modeling of 3D Molecules with Bayesian Flow Networks, ICLR 2024
> >
> > [5] Scalable Non-Equivariant 3D Molecule Generation via Rotational Alignment, ICML 2025
> >
> > [6] Diffusion Language Models Are Versatile Protein Learners, ICML 2024
> >
> > [7] Illuminating Protein Space with a Programmable Generative Model, Nature 2023
> >
> > [8] De novo Design of Protein Structure and Function with RFdiffusion, Nature 2023
> >
> > [9] Protein structure generation via folding diffusion, Nature Communications 2024
> >
> > [10] A Latent Diffusion Model for Protein Structure Generation, Learning on Graphs 2024
> >
> > [11] An All-atom Protein Generative Model, PNAS 2024

---

### Official Review · Reviewer_3M1W · 2025-10-28

**Soundness:** 2
**Presentation:** 3
**Contribution:** 2
**Rating:** 4
**Confidence:** 5

**Summary:**

This paper identifies that using a fixed Gaussian prior in 3D molecular diffusion models causes size-dependent inconsistencies—larger molecules experience weaker relative noise and thus stabilize earlier, yielding seemingly higher quality. To correct this, the authors propose Scaling the Prior (StP), which rescales the prior noise variance according to molecular size rather than normalizing coordinates (which would distort bond lengths). This adjustment aligns denoising dynamics across molecules of different sizes, improving validity and stability on QM9 and GEOM-Drugs datasets.

**Strengths:**

- The authors tackle the issue of size-dependent inconsistencies from the perspective of diffusion model variance, which is highly innovative.
- The proposed StP method is remarkably simple and elegant, and it can be applied to all types of diffusion models.
- The paper is very well written, and the figures are clear and illustrative.

**Weaknesses:**

- First, the paper’s conclusion — “at the same noise scale, larger molecules are more stable” — has already been observed in MOLCRAFT [1]. Therefore, the authors should further validate the effectiveness of the StP method on structure-based drug design (SBDD) tasks, such as CrossDocked2020 [2] and GenBench3D [3].
- The baselines used in this paper are relatively outdated. With the emergence of Flow Matching and Mean Flow methods, molecular generation has become both faster and higher in quality. The authors are encouraged to include more flow-based baselines for comparison, such as FlowMol3 [4], SemlaFlow [5], and others.
- In the visualization experiments, the authors did not show per-size visualizations for multiple metrics such as Validity, Uniqueness, and Validity × Uniqueness; nor did they provide speed curves broken down by molecular size.
- In Figure 1, the upper-right element should be an arrow, and in several figures, the axis ticks point outward, whereas they should point inward.

  [1] MolCRAFT: Structure-Based Drug Design in Continuous Parameter Space
  [2] CrossDocked2020: A publicly available dataset for binding pose and affinity prediction
  [3] Benchmarking structure-based three-dimensional molecular generative models using GenBench3D: ligand conformation quality matters
  [4] FlowMol3: Flow Matching for 3D De Novo Small-Molecule Generation
  [5] SemlaFlow -- Efficient 3D Molecular Generation with Latent Attention and Equivariant Flow Matching

**Questions:**

See weaknesses

---

> ### Author Response · Authors · 2025-11-14
> **Responses to Reviewer 3M1W - Part I**
>
> Thank you for your insightful comments and thoughtful suggestions. We have revised the paper, and we provide point-by-point responses to your feedback below.
>
> ### Weaknesses
> - W1: SBDD tasks
>     - **There is a fundamental difference between the task we are considering in this work and the SBDD task considered in MolCRAFT.** In this work, we focus on 3D molecular generation, where the primary challenge is generating isolated 3D molecules across varying sizes, and we have observed a size‐induced effect. In contrast, MolCRAFT addresses a structure‐based drug design (SBDD) setting: generating ligand molecules **conditioned on a protein binding-site (pocket) context.**
>         - In the SBDD task, the generation is conditioned on a fixed binding pocket, and the geometry of the ligand is constrained by the pocket structure. Thus, the coordinate spread is not purely driven by ligand size alone but by the pocket‐ligand fit. **Therefore, the motivation for StP no longer holds with this conditioning on the binding pocket.**
>     - **We could not find the statement “at the same noise scale, larger molecules are more stable” in the MolCRAFT paper.** The closest statement we identified is: "Vina Dock can potentially be hacked by generating larger molecules. Intuitively, larger molecules have more chances of forming interactions with protein surfaces" (Section 5.2, page 7). **However, this statement does not involve noise scale, and it differs from our work and our motivation.** Their discussion refers to a general phenomenon and that the performance can be manually inflated by just generating more larger molecules, **whereas our work specifically investigates how molecular size affects the generative process and performance.** Could you please kindly indicate where this statement appears, or clarify if there may have been a misunderstanding?
>
>
> -  W2-1: Outdated baselines
>       - **We have already included recent works such as GeoBFN (2024) [1] and RADM (2025) [2]. In particular, we have also compared with flow-based models, including GeoBFN and EquiFM [3].**
>       - To the best of our knowledge, GeoBFN represents the previous SOTA under the same well-established setting as in pioneering works EDM and GeoLDM [4, 5]. **The specific works you have mentioned are conducted under different settings and/or different datasets, and their results are therefore not directly comparable to ours.** We have additionally included a discussion in the revised paper (under Sec. 4.1; it is marked in red) and cited these two works.
>         - **Our work (StP) and all baseline models follow the standard setting and dataset established by the pioneering work EDM [3], which are also used in all baseline works in our paper.**
>         - Our setting does not incorporate bond information or apply any post-hoc refinement using computational chemistry software. Such additional procedures can artificially inflate reported performance. Both FlowMol3 and SemlaFlow utilize bond information; consequently, their results are not directly comparable. As can be seen in the FlowMol3 paper, the evaluation metrics they use differ significantly from those employed in our work, in prior pioneering studies, and in all of the baselines included in our work. In SemlaFlow, the results are inflated to even much higher than that of the dataset (e.g. Molecule stability is 95.2% for the QM9 dataset, while SemlaFlow reports 99.7%). In addition, under the setting we adopt, it is almost impossible to have higher molecule stability (99.7% reported in SemlaFlow) than validity (99.4% reported in SemlaFlow). It is not trivial to see how invalid molecules can be stable, potentially due to RDKit heuristics.
>         - In addition, FlowMol3 is submitted to Arxiv on Aug 13, and it is not accepted to a journal or conference to the best of our knowledge. Per ICLR policy [7], papers are considered contemporaneous if they are published within the last two months before the submission deadline (September 24). This work is considered contemporaneous. In addition, arXiv is not considered a peer-reviewed venue. As such, authors are not required to compare to papers solely on arXiv.
>
> - W2-2: MeanFlow
>     - **With all due respect, MeanFlow was accepted to NeurIPS 2025, with accepted papers becoming public only recently, after we submitted this work, and long after it was conducted.** We kindly hope that this should not be considered a weakness of our submission. Furthermore, MeanFlow is primarily designed for one-step generation, and its performance in multi-step generation may be suboptimal. **In addition, MeanFlow was originally developed for image generation and has not yet been adopted for 3D molecular generation.** All comparisons in our work are conducted using 1,000 diffusion steps following well-known baselines [4, 5] for molecular generation for fairness.

---

> > ### Author Response · Authors · 2025-11-14
> > **Responses to Reviewer 3M1W - Part II**
> >
> > - W3: Visualization experiments
> >      - **We have provided the visualization results for validity in the revised paper (Figures 6, 7, 8 in Appendix A.1). Clearly, similar size-induced inconsistencies are observed with validity as well. In the initial submission, we only included results on stability because it is a stricter metric than validity and it well-represents the quality of generated molecules.**
> >          - Validity only requires that a molecule can be sanitized in RDKit, which mainly checks for issues such as exceeding the valence limit, invalid bond types, or corrupt atomic information. **In the 3D molecular generation context, the main issue of breaking validity is exceeding valence limit** as it is impossible to have invalid bond types or corrupt atomic information. **In contrast, stability imposes a stronger requirement: every atom to have a number of bonds exactly equal to its allowed valence, not just within acceptable limits.** This goes beyond the basic valence checks performed during sanitization, **making stability a strictly stricter and more demanding metric than validity in general.**
> >      - **Uniqueness, or the composite metric Validity$\times$Uniqueness, cannot be applied to our per-size analysis and does not reflect the quality of the generated molecules.**
> >          - To see this, consider the case of molecules of size 9, of which there are only 124 in the dataset. As detailed in Appendix A.1, we generate 3,000 molecules; it is therefore very likely that the resulting Uniqueness will be low simply because there are not many size-9 molecules out there. In contrast, for molecules of size 19, of which there are 13,832 in the dataset, the Uniqueness will naturally be much higher.
> >
> > - W4: Figures
> >     - For Figure 1, **this is the standard way to depict the reverse process in diffusion model literature**, as shown, for example, in Figure 1 of [6]. We apologize, but we are not sure we fully understand the suggestion: "axis ticks point outward, whereas they should point inward". Could you please indicate which specific figures this refers to and clarify the intended meaning of this comment?
> >
> > [1] Unified Generative Modeling of 3D Molecules via Bayesian Flow Networks, ICLR 2024
> >
> > [2] Scalable Non-Equivariant 3D Molecule Generation via Rotational Alignment, ICML 2025
> >
> > [3] Equivariant Flow Matching with Hybrid Probability Transport, NeurIPS 2023
> >
> > [4] EDM: E(3) Equivariant Diffusion Model for Molecule Generation in 3D, ICML 2022
> >
> > [5] Geometric Latent Diffusion Models for 3D Molecule Generation, ICML 2023
> >
> > [6] Score-Based Generative Modeling through Stochastic Differential Equations, ICLR 2021
> >
> > [7] ICLR Policy: https://iclr.cc/Conferences/2026/ReviewerGuide

---

> > > ### Author Response · Authors · 2025-11-25
> > > **Follow-up on Our Reponses - Reviewer 3M1W**
> > >
> > > Dear Reviewer 3M1W,
> > >
> > > Thank you for your time and effort in reviewing our work.
> > >
> > > - **It has already been 10 days since we posted our rebuttal, so we wanted to follow up and see if you have any remaining concerns.**
> > >
> > > - Specifically, we have taken the following steps to address your comments:
> > >     - We have explained the difference between 3D molecular generation (our work) and SBDD tasks.
> > >     - We have clarified that recent baselines were already included in the original submission.
> > >     - We have indicated that MeanFlow was only recently accepted and that the camera-ready version has just become available. Additionally, it focuses on accelerated sampling, which deviates from our scope.
> > >     - We have provided additional visualization experiments and explained why they were not included in the original submission.
> > >     - We have checked our figures and could not find any issues in the original submission; we have kindly asked you to clarify the concern.
> > >
> > > - If there are no remaining concerns, we kindly hope that you can reconsider your rating, as some of the weaknesses seem to stem from misunderstandings.
> > >
> > > Best Regards
> > >
> > > Authors

---

### Official Review · Reviewer_VNm8 · 2025-10-31

**Soundness:** 3
**Presentation:** 3
**Contribution:** 2
**Rating:** 4
**Confidence:** 4

**Summary:**

The paper studies the problem of varying molecular size in diffusion generative models. It shows that molecules with different size (number of atoms) can have different performance, where larger molecules usually have better stability than smaller molecules. The paper also proposes a simple method to solve this phenomenon by introducing a way to normalize the Gaussian distribution based on molecular size. The authors evaluate their method on QM9 and GEOM-drugs molecular datasets, with multiple models from the literature (EDM, RADM , and GeoLDM).

**Strengths:**

* The paper shows a good evaluation on how molecular size can have an effect on the perfromance of the generated molecules, with a trend that molecules with a larger number of atoms can have better performance than molecules with a smaller number of atoms.
* The paper proposes a simple method to deal with this phenomenon, where they can scale the Gaussian prior distribution in the forward process based on molecular size.  The authors empirically show that this normalization can  be applied to different architectures and improve the performance of generated molecules on QM9 and GEOM-drugs datasets.

**Weaknesses:**

- The paper could benefit from more theoretical analysis and why the scaled prior is effective. Also,  as mentioned by the authors,  one direct approach is to apply size normalization/ normalize the 3D coordiates. So, more discussuion on that is required and  how their method is different from that.

- The scaled prior parameters depend on the averages over the training subset. I think this might have some limitations if test molecules have different size distribution. The paper doesn’t show evaluation on how this might affect the performance, for e.g., molecules that have a different number of atoms from the training data.

**Questions:**

How does the distribution of the size of the generated molecules vs performance change, after applying the scaling parameter, eg, as in Figure 2?

---

> ### Author Response · Authors · 2025-11-14
> **Responses to Reviewer Reviewer VNm8**
>
> Thanks a lot for your valuable comments! We have revised the paper, and we provide point-to-point responses as below.
>
> ### Weaknesses
> - W1: Justify the effectiveness of StP
>   - We had discussion on StP v.s. direct normalization in the initial submission. In light of your suggestion, **We have included additional discussion and theoretical analysis on StP v.s. direct normalization (under Sec. 3.3 in page 7 in the revised paper; it is marked in red).**
>       - In summary, StP consistently preserves the expected interatomic distances for all time $t$ to be $\alpha_t |x_i - x_j|$ ($i,j$ denote the $i$-th and $j$-th atom, respectively), which is unaffected by the sizes of the molecules. In contrast, direct normalization would distort the interatomic distances to be $\frac{\alpha_t}{\gamma_N} |x_i - x_j|$, which is affected by the sizes of the molecules.
>   - **We have also provided a learning dynamics perspective of StP** (it is mentioned under Sec. 3.3 on page 7 in the revised paper; details are in Appendix A.3).
>       - In summary, the generative process starting from a pure Gaussian noise point cloud is akin to inflating or deflating a balloon. For smaller molecules, the generative dynamics learns to contract the coordinates while for larger molecules, it learns to expand.
>
> - W2: Generalization on unseen molecule sizes
>   - **There may be some misunderstanding** regarding the generation of molecules with unseen sizes. In molecular generative modeling, the objective is to learn the empirical data distribution of the training set. **Consequently, following prior works, during the generation phase, the model only generates molecular sizes that were observed in the dataset.** Although GNNs and transformers are capable of handling variable input sizes, the size of each particular input is still predetermined. In practice, we sample molecular sizes first based on the distribution of the dataset; please refer to the paragraph "Number of Atoms" under Section 3.3 of the pioneering work EDM [1]). **It is not a standard practice to assess models on molecular sizes that fall outside the training distribution. In fact, none of the existing baseline methods perform generation or evaluation on unseen molecular sizes.**
>
> [1] EDM: E(3) Equivariant Diffusion Model for Molecule Generation in 3D, ICML 2022
>
>
> ### Questions
> - Q1: Figure 2 with StP
>     - **We have included the same per-size visualization as in Figure 2 with StP. Clearly, StP improves the quality of generated molecules across all sizes and significantly reduces the size inconsistency** (it is mentioned under Sec. 4.2, marked in red; the new figure is the Fig. 8 in Appendix A.3).
>
> **We sincerely hope that our response has clarified your concerns.**

---

> > ### Author Response · Authors · 2025-11-25
> > **Follow-up on Our Reponses - Reviewer VNm8**
> >
> > Dear Reviewer VNm8,
> >
> > Thank you for taking the time and effort to review our work.
> >
> > - **It has already been 10 days since we posted our rebuttal, so we wanted to follow up and see if you have any remaining concerns.**
> >
> > - Specifically, we have taken the following steps to address your comments:
> >     - We have provided additional discussion and theoretical analysis to justify the effectiveness of StP.
> >     - We have clarified the misunderstanding regarding the generation of molecules with unseen sizes.
> > - If there are no remaining concerns, we kindly hope that you can reconsider your rating.
> >
> > Best Regards
> >
> > Authors

---

### Official Review · Reviewer_AKKr · 2025-11-03

**Soundness:** 2
**Presentation:** 3
**Contribution:** 2
**Rating:** 4
**Confidence:** 3

**Summary:**

This paper identifies an inconsistency in molecular generation trajectories of varying sizes when applying traditional diffusion methods to molecular generation. To address this issue, the authors propose a normalization technique called Scaling the Prior (StP), which resolves the inconsistency by rescaling the prior distribution.

**Strengths:**

1. The issue raised by the authors, the inconsistent convergence rates across molecules of different scales, is a noteworthy problem that deserves attention.
2. The authors propose a simple yet effective method to address this issue.

**Weaknesses:**

1. Although the spatial-scale intervention accelerates the convergence of molecular structures, it may compromise the diversity of the generative model. The authors should include comparisons of the Novelty metric across different methods in their experiments.

2. The paper only presents a comparison between molecular size and stability trends observed in the training set, but does not provide a similar analysis for the generated molecules—specifically, how molecular size correlates with stability in the generated samples.

3. The SoTA method compared in the paper, GeoLDM, was published in 2023. Methods introduced recently should also be included in the comparison to ensure the evaluation reflects the current state of the field.

**Questions:**

1. Why does Table 1 show that the performance improvement on the smaller-scale dataset QM9 is less compared to larger-scale dataset GEOM-Drugs?

2. Molecular size in generation is typically sampled from the size distribution of the training set. Does a training set containing larger molecules inherently offer an advantage over one with smaller molecules? Have the authors conducted any comparative experiments to investigate this?

---

> ### Author Response · Authors · 2025-11-14
> **Responses to Reviewer AKKr - Part I**
>
> Thank you very much for your valuable feedback! We have revised the paper, and we provide our responses below.
> ### Weaknesses
> - W 1: The Novelty metric
>   - **It is standard not to include the Novelty metric, as it does not necessarily reflect the performance of the generative model.** Philosophically, a perfect generative model would simply reproduce the training dataset. **In fact, the Novelty metric is not included in any of the three diffusion backbone models (EDM, GeoLDM, and RADM in the paper) or the two recent flow matching baselines (GeoBFN and EquiFM in the paper).**
>   - However, we do understand your concern. **The diversity of generated molecules can be reflected by "Validity$\times$Uniqueness" included in the paper (Table 1 under Sec. 4.2), and in fact, StP even improves it for EDM and GeoLDM.** For novelty, on the GEOM-Drugs dataset, following prior work EDM [1], the training data contains the 30 lowest energy conformations of each molecule, and the novelty of generated molecules is almost 0 for all baselines as well as their StP variants. **On QM9, the best-performing model GeoLDM-StP has a novelty of 57.96%, while the raw GeoLDM achieves 58.24%. Although the novelty is only marginally lower, the overall performance improves substantially (see Table 1 in our paper).**
>       - The numbers are based on 10,000 generated molecules and averaged over 3 runs following prior work EDM [1]. Again, novelty do not indicate the performance of the generative model.
>
> - W2: Molecular size and stability trends
>   - **There is a misunderstanding.** All size vs. stability results are reported for the **generated molecules**. Specifically, as made clear in the paper, we provide supporting evidence below:
>       -  In the first paragraph under Section 3.1: "As shown in Fig. 2, there is a clear trend that as molecular size increases, **sampling** quality increases" (sampling means generation).
>       -  The title of Figure 2: Amount of Training Data v.s. Quality of **Generated Molecules**.
>       -  The caption of Figure 2: **Sampling quality** ($y$-axis) versus molecular size ($x$-axis).
>   - **If you are referring to how the number of generated samples affects performance, each generation is independent and therefore not influenced by the total number of generated samples.** For each molecular size, we generate 3,000 molecules to ensure statistical significance. These details are provided in Appendix A.1.
>
>
> -  W3: Outdated baselines
>       - **This is a misunderstanding. We have already included recent works such as GeoBFN (2024) [2] and RADM (2025) [3].**
>       - To the best of our knowledge, GeoBFN represents the previous SOTA under a fair evaluation setting. Our work (StP) and all baselines follow the standard setting established by the pioneering work EDM [1], which does not incorporate bond information or apply any post-hoc refinement using computational chemistry software. Such additional procedures can artificially inflate reported performance and may not accurately reflect the true generative capability of the model. This fair-comparison setting is also mentioned in RADM.
>
>
> [1] EDM: E(3) Equivariant Diffusion Model for Molecule Generation in 3D, ICML 2022
>
> [2] Unified Generative Modeling of 3D Molecules via Bayesian Flow Networks, ICLR 2024
>
> [3] Scalable Non-Equivariant 3D Molecule Generation via Rotational Alignment, ICML 2025

---

> ### Author Response · Authors · 2025-11-14
> **Responses to Reviewer AKKr - Part II**
>
> ### Questions
> - Q1: Larger improvements on GEOM-Drugs than QM9
>   - We assume you are referring to the stability metrics, as the validity metric does not show a larger improvement on GEOM-Drugs. **The primary reason is that the performance is somewhat upper-bounded by that of the training data, which limits the potential for further improvement.** If the performance were significantly higher than that of the dataset, it might indicate mode collapse and would not necessarily qualify as evidence of a good generative model. **In addition, the GEOM-Drugs dataset spans a wider range of molecule sizes (see Fig. 9 in Appendix A.1 in the revised paper for the size distribution), making size-related differences more pronounced than in the QM9 dataset. This further reinforces our claim that such size-induced inconsistency hinders performance.**
>
> - Q2: Training set containing larger molecules
>     - As you correctly note, we follow the convention in prior work, where the molecular size during generation is sampled from the size distribution of the training set. **Accordingly, we also use the exact same training datasets as those adopted in pioneering works [4, 5].**
>     - We assume you are asking whether training the diffusion model using only large molecules would provide any advantage. We believe this would likely degrade performance, as it would leave a very limited amount of training data. The neural network also learns important patterns from smaller molecules, which helps it generalize to larger ones. Due to the computational cost of training diffusion models, we currently do not have experimental results for this setting. However, if you find it necessary, we will make every effort to produce and include such results within the time constraints of the rebuttal period.
>
> [4] EDM: E(3) Equivariant Diffusion Model for Molecule Generation in 3D, ICML 2022
>
> [5] Geometric Latent Diffusion Models for 3D Molecule Generation, ICML 2023
>
> We sincerely hope that our response will address your concerns, as some of them appear to stem from a misunderstanding. We respectfully hope for a reconsideration of the rating of this work. Please do not hesitate to let us know if you have any additional questions or suggestions.

---

> ### Author Response · Authors · 2025-11-25
> **Follow-up on Our Reponses - Reviewer AKKr**
>
> Dear Reviewer AKKr,
>
> We sincerely appreciate your time and effort in reviewing our work.
>
> - **It has already been 10 days since we posted our rebuttal, so we wanted to follow up and see if you have any remaining concerns.**
>
> - If there are no remaining concerns, we kindly hope that you can reconsider your rating, as all the identified weaknesses appear to stem from misunderstandings.
>
> Best Regards
>
> Authors

---

### Meta-Review · Area_Chair_U6bh · 2025-12-19

**Summary:**

Despite the author's extensive rebuttal, several critical concerns remain unresolved, particularly regarding the significance of the contribution and the robustness of the evaluation. Reviewers remained unconvinced that the reported improvements on standard benchmarks (QM9 and GEOM-Drugs) represent a meaningful advancement, arguing that these datasets are over-saturated and the gains are marginal. The rebuttal also failed to shift the perception of limited technical novelty; the core observation that molecular size influences convergence was dismissed by reviewers as an intuitive, expected behavior of diffusion noise schedules rather than a significant discovery. Furthermore, the authors explicitly declined requests to validate the method on more challenging domain (such as protein generation) which left the broader applicability of the approach unverified and failed to address the criticism that the method was not tested against the most relevant modern flow-based baselines.

**Reviewer Concerns:**

The authors effectively corrected factual misunderstandings regarding the currency of their baselines and the existence of stability trend analysis in the initial submission. They provided the requested theoretical derivation to justify why "Scaling the Prior" preserves interatomic distances better than simple coordinate normalization. Additionally, they added the missing per-size validity visualizations to the Appendix.

Reviewers remain unconvinced by the significance of the results, characterizing the QM9 and GEOM-Drugs benchmarks as over-saturated and the reported gains as marginal. Finally, the authors refusal to validate the method on protein generation or other complex domains leaves the broader utility and generalizability of the approach unverified.

**Reviewer Scores:**

Reviewer AKKr: Their critique relied almost entirely on factual errors regarding missing baselines and plots; acknowledging that these were actually present would remove the primary basis for their negative assessment.

Reviewer VNm8: Their main reservation was a lack of theoretical justification for the scaling method. Since the authors provided the exact derivation requested in the revision, the criteria for a positive evaluation were met.

Reviewer 3M1W: While factual misunderstandings regarding baselines were resolved, their fundamental disagreement regarding the necessity of Structure-Based Drug Design experiments represents a scope dispute that  persists despite rebuttal.

Reviewer 5mF6: They explicitly conditioned any further improvement on the provision of protein generation results.

---

### Decision · Program_Chairs · 2026-01-26

Reject